# VIDEOGAMEBENCH: CAN VISION-LANGUAGE MODELS COMPLETE POPULAR VIDEO GAMES?

## ABSTRACT

Vision-language models (VLMs) have achieved strong results on coding and math benchmarks that are challenging for humans, yet their ability to perform tasks that come naturally to humans–such as perception, spatial navigation, and memory management–remains understudied. Real video games are crafted to be intuitive for humans to learn and master by leveraging innate inductive biases, making them an ideal testbed for evaluating such capabilities in VLMs. To this end, we introduce VideoGameBench, a benchmark consisting of 10 popular video games from the 1990s that VLMs directly interact with in real-time. VideoGameBench challenges models to complete entire games with access to only raw visual inputs and a high-level description of objectives and controls, a significant departure from existing setups that rely on game-specific scaffolding and auxiliary information. We keep three of the games secret to encourage solutions that generalize to unseen environments. Our experiments show that frontier vision-language models struggle to progress beyond the beginning of each game. We find inference latency to be a major limitation of frontier models in the real-time setting; therefore, we introduce VideoGameBench Lite, a setting where the game pauses while waiting for the LM's next action. The best performing models, Gemini 2.5 Pro and Claude 3.7 Sonnet, complete only **0.48%** of VideoGameBench and **1.6%** of VideoGameBench Lite. We hope that the formalization of the human skills mentioned above into this benchmark motivates progress in these research directions.

## 1 INTRODUCTION

Language Models (LMs) and vision-language models (VLMs) perform complex tasks remarkably well, even those that are challenging to humans such as advanced mathematics (Azerbayev et al., 2024; Lin et al., 2025) and coding (Li et al., 2022; Luo et al., 2025; OpenAI et al., 2025). However, that does not necessarily mean that they demonstrate human-level performance on all tasks. Humans have perceptual, spatial, and memory management abilities that provide strong inductive biases for learning new tasks (Lake et al., 2017; Dubey et al., 2018). To evaluate whether current AI systems are approaching those abilities, we propose a new challenge: completing video games from the 1990s (also known as the 32-bit era).

We introduce VideoGameBench, a benchmark which challenges VLMs to complete, in real-time, a suite of 10 different popular video games from both hand-held consoles (Game Boy and Game Boy Color) and PC (Microsoft DOS). Solving video games relies on fundamental multi-modal reasoning abilities (Shao et al., 2019)—e.g. spatial awareness, memory retention, efficient exploration strategies, and real-time reaction to dynamic events. Video games are carefully crafted to be learnable and playable by humans, catering to human inductive biases (Allen et al., 2024). As a result, they provide an ideal setting for exploring how well agents reproduce those inductive biases (Dubey et al., 2018).

VideoGameBench has three important novel features:

1. It challenges VLMs with significantly more complex and realistic environments than those found in earlier benchmarks, such as grid-world or text-only games (Paglieri et al., 2025; Nasir et al., 2024), and is one of the first benchmarks to use video games from the 1990s.

2. It evaluates how a single agent performs across different games, including three secret games specifically designed to test generalization to unseen or out-of-distribution scenarios. Unlike

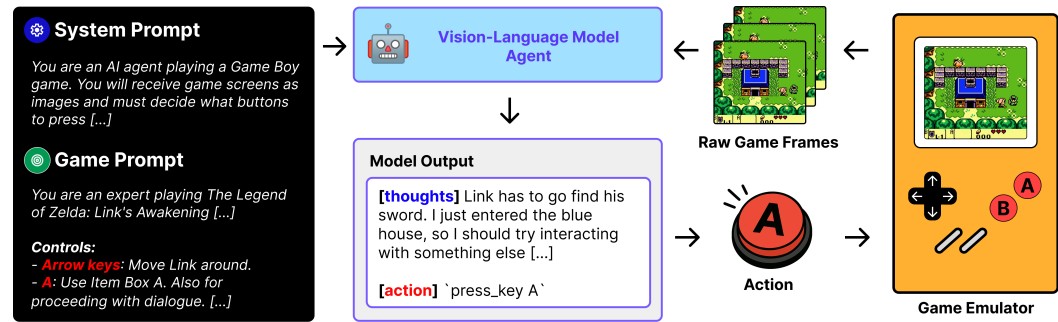

Figure 1: VideoGameBench provides an environment for vision-language models (VLMs) to interact with video game emulators – for example, the emulator above is playing *The Legend of Zelda: Link's Awakening*. Given information about the game controls and emulator and access to the game's raw frames, models provide actions in natural language.

previous works (Mnih et al., 2013; OpenAI et al., 2019; Vinyals et al., 2019; Rubinstein et al., 2025), it challenges agents with environments that they may not have been trained on.

3. It only provides agents with raw game visual inputs, and does not allow game-specific hints, visual overlays or tools (Kempka et al., 2016; Hershey, 2025). Recently, *Gemini Plays Pokemon* (Joel Z, 2025) showed that a frontier VLM (Google DeepMind, 2025), with tailored tools for pathfinding, game-specific hints, and memory, could complete *Pokemon Blue*. Although VideoGameBench includes similar games, we focus on evaluating VLMs with minimal human or tool-assisted intervention.

We evaluate multiple frontier VLMs on VideoGameBench using our VG-Agent scaffolding and find that all models struggle to progress on any game – the best performing models, Gemini 2.5 Pro (Google DeepMind, 2025) and Claude Sonnet 3.7 (Anthropic, 2025), are able to achieve a score of **0.48%** on VideoGameBench, which represents the average amount of each game that the agent completes. Our VG-Agent uses ReAct (Yao et al., 2023) with the ability to store information in context (Shinn et al., 2023), and has basic information on the controls and objectives for each game. We also introduce a set of simpler practice games to evaluate skills such as mouse movement and navigation and find that most models perform poorly. Finally, to enable more granular progress tracking, we release VideoGameBench Lite–a smaller benchmark where the emulator pauses during agent inference, eliminating latency issues from slow model responses.

## 2 VIDEOGAMEBENCH

VideoGameBench is a benchmark composed of a diverse suite of 23 curated video games split across a `dev` and `test` set, with an environment to evaluate and communicate with VLM-based agents. The task is to solve the *core objective* of each game, e.g. defeating the final boss in *Super Mario Land* or completing the entire single-player campaign for *Age of Empires*. We explicitly provide a `dev` set to encourage improving the performance of models or minimal scaffolds on interactive environments without using the `test` set games.

### 2.1 BENCHMARK CONSTRUCTION

To enable our benchmark to run games from multiple different platforms in a modern computer environment, our VideoGameBench framework abstracts the underlying game emulator (currently supporting Game Boy via `PyBoy` (Mads Ynddal, 2025) and MS-DOS games via `DOSBox` (Veenstra et al., 2002), `JS-DOS` (Guryanov, 2025), and `Playwright` (Microsoft, 2020)) and provides a standardized interface for agents to communicate with.

The focus on Game Boy, Game Boy Color, and classic MS-DOS video games was motivated by:

1. **2D / 3D Environments:** Compared to previous benchmarks (Paglieri et al., 2025; Nasir et al., 2024; Waytowich et al., 2024), the VideoGameBench games introduce more realistic and challenging

visuals and also require significant planning (e.g. navigating the world in *The Legend of Zelda: Link's Awakening*), puzzle-solving (e.g. understanding physics in *The Incredible Machine*), and strategic thinking (e.g. dealing with multiple enemies shooting in *Doom II*).

2. **Diverse Mechanics:** Our game selection covers both controller-based mechanics (Game Boy) and mouse/keyboard interactions (MS-DOS), presenting different challenges for agent control. For example, mouse control requires precisely mapping points on a fixed screen, while controller D-Pad (the flat, four-way directional button on a controller) movement requires understanding positions relative to the game character.

## 2.2 TASK FORMULATION

In VideoGameBench, VLM agents are provided instructions on emitting game actions (e.g. press "A" to jump), and the raw game screen without extra information as input, and provide an action as output every step. We do not permit human-written hints beyond high-level game objectives and a description of game controls (see Appendix B).

**Agent inputs.** The primary observation provided to the agent at every step is the raw game screen as an image (or set of frames). We deliberately avoid providing parsed text overlays or structured state information (as in previous works including PySC2 (Vinyals et al., 2017) and VizDoom (Kempka et al., 2016)) to challenge the model's visual processing capabilities. In addition, we do not provide the agent with any intermediate rewards or signals throughout gameplay.

**Agent actions.** The agent interacts with the game controller through a language-based action interface. This supports single actions (e.g., press "Space"), sequences of actions (e.g., press "A", "A", "Start"), or sequences of timed actions (e.g., hold "Up" for 2 seconds). For example, the agent outputs "`press_key space`" to press the space bar for a default of $1/10$ a second. The specific available actions depend on the game platform (keyboard/mouse for MS-DOS, button presses for Game Boy).

**Scoring.** Each game is weighted equally (out of 1) and is divided uniformly by the number of predefined checkpoints on each game. Checkpoints were manually defined based on well-known level or stage divisions for these games (see Appendix C.4). Furthermore, we specifically allow agents to restart games after losing to allow agents to adjust their strategies. To prevent running infinitely long trajectories, we provide a list of specifications to determine if a run ends by capping the overall play-time per game (see Appendix A).

**VideoGameBench Rules.** There are many possible approaches to playing video games using a VLM agent. For example, one could allow engineering visual overlays on top of the raw game screen to assist the agent. In addition, a benchmark designer could allow providing the agent with tools that access internal game-state information such as the game emulator's internal RAM.

On VideoGameBench, we want to evaluate a VLM agent's ability to interpret raw visual inputs and directly interact with the game environments to achieve its goals. Therefore, we do not permit any visual overlays, and also do not permit providing any extra information to the agent sourced either from the internal game-state or from the viewable screen. We only allow the raw game frames and a basic description of the game's objectives and controls as input to the agent.

## 2.3 THE GAMES OF VIDEOGAMEBENCH

VideoGameBench includes 10 games in the `test` set, and 13 in the `dev` set, described in Table 1. Previous game benchmarks for VLMs such as Paglieri et al. (2025); Nasir et al. (2024); Tsai et al. (2025) all include six or less games, so we limit our test set to 10 games due to the difficulty, length, and diversity of each game. Seven of these are public and three are secret (see §2.4).

**Game Genres.** The VideoGameBench games were selected based on popular genres of video games, each offering unique and unique challenges. We highlight challenges we observed for agents below:

- **First-person shooter (FPS)**: Agents must traverse 3D environments, aim at enemies and dodge enemy attacks.

- **Platformer**: Agents must move a character under 2D physics mechanics to reach a goal, all while fighting and avoiding obstacles.

Table 1: VideoGameBench games and their properties. Three of the games in our test set are kept secret. The right-most column indicates if the video game is "real-time", i.e. it requires real-time actions. (✓) means it requires real-time responses. (✗) indicates that it does not require real-time responses. The Lite subset effectively removes this constraint. DX indicates the colored version of an originally Game Boy game.

| | Game | Genre | Dim. | Platform | Real-time? |
|---|---|---|---|---|---|
| **Development Set** | Quake | FPS | **3D** | MS-DOS | ✓ |
| | Prince of Persia | Platformer/Action | 2D | MS-DOS | ✓ |
| | Super Mario Land | Platformer | 2D | Game Boy | ✓ |
| | Doom | FPS | **3D** | MS-DOS | ✓ |
| | Warcraft II | Strategy (Real-time) | 2D | MS-DOS | ✓ |
| | The Oregon Trail Deluxe | Simulation/Educational | 2D | MS-DOS | ✓ |
| | X-COM: UFO Defense | Strategy (TBS/Tactical) | 2D | MS-DOS | ✓ |
| | Scooby-Doo: Classic Creep Capers | Adventure | 2D | GB Color | ✓ |
| | Age of Empires | Strategy (Real-time) | 2D | MS-DOS | ✓ |
| | Pokémon Red | RPG | 2D | Game Boy | ✗ |
| | Castlevania: The Adventure | Platformer/Action | 2D | Game Boy | ✓ |
| | Donkey Kong Land 2 | Platformer | 2D | Game Boy | ✓ |
| | Mega Man: Dr. Wily's Revenge | Platformer/Action | 2D | Game Boy | ✓ |
| **Test Set** | Doom II | FPS | **3D** | MS-DOS | ✓ |
| | Kirby's Dream Land (DX) | Platformer | 2D | Game Boy | ✓ |
| | Zelda: Link's Awakening (DX) | Action-Adventure/RPG | 2D | Game Boy | ✓ |
| | Sid Meier's Civilization | Strategy (Turn-based) | 2D | MS-DOS | ✗ |
| | The Need for Speed | Racing | **3D** | MS-DOS | ✓ |
| | The Incredible Machine | Puzzle | 2D | MS-DOS | ✗ |
| | Pokémon Crystal | RPG | 2D | GB Color | ✗ |
| | Secret Game #1 | ? | ? | ? | ? |
| | Secret Game #2 | ? | ? | ? | ? |
| | Secret Game #3 | ? | ? | ? | ? |

- **Action-Adventure / RPG**: Agents must reason over complex strategies to defeat or ally with enemies, and must keep track of long-horizon game state information such as their own resources and the current objectives.

- **Racing**: Agents must continuously adjust and react in a 3D vehicle environment and race against other opponents.

- **Puzzle**: Agents must solve a series of puzzles based on in-game physics mechanics and tools.

## 2.4 SECRET GAMES

Our objective is not to measure whether agents can play *specific* games, but whether they can adapt to, learn from, and progress in new environments. Therefore, we also test each agent on three secret games that we host on a private evaluation server. We determined that these games are of similar difficulty to the other games in VideoGameBench, each falling under one of the genres described in § 2.3. We believe that including these games could incentivize agent researchers to focus on systems that can generalize to new environments, rather than focusing on narrowly-developed agents that can only play a limited, pre-determined set of games.

## 2.5 VIDEOGAMEBENCH LITE: GIVING AGENTS TIME TO THINK

A major bottleneck for current VLMs in real-time games is inference latency. We observe that VLMs take so much time to respond that by the time they return an action to perform, the game state has already substantially changed, so the action choice the agent made is now stale. When using an agent that processes multiple historical frames along with the current game image, along with a history of previous actions and thoughts, processing times increase, further exacerbating this issue.

To disentangle reasoning ability from reaction time constraints, we introduce **VideoGameBench Lite**, a variant of the full benchmark where the underlying emulator is paused whenever the agent is processing information and deciding on its next action – the game only resumes momentarily when an action is made. This change effectively turns the real-time game into a turn-based interaction,

allowing evaluation of the agent's planning and decision-making capabilities irrespective of inference speed. The list of VideoGameBench Lite games is: *Quake*, *Prince of Persia*, *Super Mario Land* from the `dev` set, and *Doom II*, *Kirby's Dream Land*, *Zelda: Link's Awakening* from the `test` set.

## 2.6 Automated Progress Tracking

To make VideoGameBench finer-grained, and not just one in which we assign binary pass/fail scores to each game, we introduce an automated progress tracking mechanism to detect what levels or stages of the game the agent managed to complete in each run. To do this, we introduce a novel method for tracking progress: we first scrape game screen checkpoint images (e.g. the end of a level) from YouTube video walkthroughs of each game – specifically walkthroughs that have timestamp pointers for each level in their descriptions (see Appendix C.3). We show the distribution of these checkpoints throughout each game in Appendix C.6.

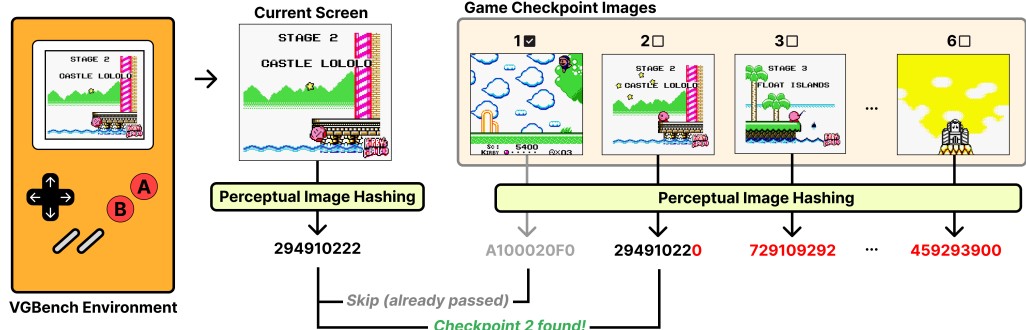

Figure 2: To track progress on VideoGameBench, we scrape deterministic checkpoints from online walkthroughs of video games and compute image hashes using (Buchner, 2013). These hashes are compared to the current game screen to determine if an agent has reached a checkpoint, and the score is determined based on the relative timestamp of the checkpoint with respect to the walkthrough.

We automatically detect which checkpoint an agent has reached by applying perceptual image hashing (Marr & Hildreth, 1980; Buchner, 2013) to every frame outputted by the emulator during the agent's gameplay. We compute the hamming distance (Hamming, 1950) between the current game frame and all of the perceptual hashes of the scraped checkpoint frames and check if they have a hamming distance below a certain threshold (see Appendix C.5) to determine if an agent has reached a certain checkpoint of the game (see Figure 2). We note that our frame matching method is similar to Toufool et al. (2025), which is used in the "speedrunning" community; however, their system is meant to be used as standalone software that does not easily plug into our system. To compute the agent's VideoGameBench score for a specific game, we track the furthest checkpoint reached by the agent, lookup the timestamp in the associated walkthrough video, and compute the percentage of the game completed. See Appendix C.4 for a detailed list of these manually determined checkpoints.

## 3 Experimental Setup

In this section, we describe the inputs provided to agents running on VideoGameBench. We also explain our agent baselines that we use to evaluate the current performance on this benchmark.

**VG-Agent.** Because the games in VideoGameBench require context that is often not visible on-screen, the naïve approach of only providing a VLM the current game screen would fail. Therefore, we implement a ReACT (Yao et al., 2023) agent to play video games. This agent is given (1) an initial prompt with basic information about the game objectives as our baseline evaluation. The agent is given a single frame after half a second of taking an action, and each action lasts for a fraction of a second (e.g. "press_key A" presses the "A" key for 0.1 seconds), (2) an initial prompt with explicit game control instructions such as what the "A" button does, and (3) a prompt that asks the model to output which information it wants to store as a textual scratchpad (memory) after every step. We provide the last few frames and agent observations in context.

**Models.** VideoGameBench requires interpreting raw game screenshots and producing structured outputs in language while also understanding the game. We evaluate exclusively on VLM models, namely GPT-4o (`gpt-4o-2024-08-06`) (OpenAI et al., 2024), Claude Sonnet 3.7 (`claude-3-7-sonnet-20250219`) (Anthropic, 2025), and Gemini 2.5 Pro (`gemini-2.5-pro-preview-03-25`) (Google DeepMind, 2025), three of the leading closed-source VLMs. We also evaluate Gemini 2.0 Flash (`gemini-2.0-flash`) (Google, 2025) for its fast inference and Llama 4 Maverick (`Llama-4-Maverick-17B-128E-Instruct-FP8`) (Meta AI, 2025; Together AI, 2025), and the 7B and 32B variants of QwenVL2.5 (Bai et al., 2025) to evaluate prevalent open source models across different scales. Following Yao et al. (2023), all models are run with temperature 0.7 and a maximum output length of 2048 tokens.

**Constraints.** There are many games in which the model could run infinitely long because it gets stuck without ending the game. Given a limited budget, we preemptively end runs based on rules indicating insufficient progress toward the next checkpoint (see Appendix D.1).

**Ensuring the validity of our interface.** To validate the completeness of our interface, we had a human player complete the first levels of a few games using only the same information available to the language model. For details on the human interface, see Appendix C.2.

**VG-Agent on VideoGameBench Lite.** The setup on VideoGameBench Lite modifies the simulator to pause while the agent processes the input to generate an action, similar to the OpenAI Gym environment (Brockman et al., 2016). For our experiments, at every step on the Game Boy emulator we provide the agent with the most recent game frame, while on the DOS emulator we provide it with five past frames spaced 0.1 seconds apart. Other than this, we evaluate all models with the same settings as VideoGameBench, including the same game-specific system prompts.

## 4 RESULTS

We evaluate VG-Agent on the `test` split of VideoGameBench and VideoGameBench Lite across different frontier VLMs in Table 2, then provide analysis on model's performance and their failure modes. The best-performing models, Gemini 2.5 Pro and Claude Sonnet 3.7, are only able to reach the first checkpoint in a single game. Due to budget constraints, we limit each model to one run on each game. Our study of system variance (on three games using Gemini 2.5 Pro) shows that there is little variance in these settings; therefore, we believe that running these experiments multiple times would not have led to substantially different results (see Appendix D.3).

Table 2: Performance on VideoGameBench `test` split, a benchmark consisting of 7 games and 3 secret games, which we keep private to users. Each score is displayed as as a percentage of the game completed based on completed checkpoints, i.e. 0% means the agent did not reach the first checkpoint. The overall score is computed as an average of scores across all games. The cost of each run is reported in Appendix D.2.

| VideoGameBench | GPT-4o | Sonnet 3.7 | Gemini 2.5 Pro | LLaMA 4 | Gemini 2.0 Flash | QwenVL-2.5 7B | QwenVL-2.5 32B |
|---|---|---|---|---|---|---|---|
| Civilization I | 0% | 0% | 0% | 0% | 0% | 0% | 0% |
| The Need for Speed | 0% | 0% | 0% | 0% | 0% | 0% | 0% |
| The Incredible Machine | 0% | 0% | 0% | 0% | 0% | 0% | 0% |
| Pokemon Crystal | **0.9%** | 0% | 0% | 0% | 0% | 0% | 0% |
| Doom II | 0% | 0% | 0% | 0% | 0% | 0% | 0% |
| Kirby's Dream Land (DX) | 0% | **4.8%** | **4.8%** | 0% | 0% | 0% | 0% |
| Link's Awakening (DX) | 0% | 0% | 0% | 0% | 0% | 0% | 0% |
| Secret Game #1 | 0% | 0% | 0% | 0% | 0% | 0% | 0% |
| Secret Game #2 | 0% | 0% | 0% | 0% | 0% | 0% | 0% |
| Secret Game #3 | 0% | 0% | 0% | 0% | 0% | 0% | 0% |
| Overall Score | 0.09% | **0.48%** | **0.48%** | 0% | 0% | 0% | 0% |

### 4.1 BENCHMARK PERFORMANCE

In Table 2, we show similar trends among all models, which struggle to make progress in any game – Gemini 2.5 Pro and Claude Sonnet 3.7 achieve the highest score of **0.48%** on VideoGameBench.

**Interpreting "0%" scores on games.** VideoGameBench's scoring scheme provides a score for reaching significant game checkpoints, so a score of "0%" does not imply that the model did not progress at all (see Table 5 in Appendix C.4). Similar to Hershey (2025) and Tsai et al. (2025), we

track or provide rewards based on well-defined game milestones. As described in Section 2.6, we estimate completion percentages based on how far along a scraped walkthrough the checkpoint frame appeared. Due to the low scores on each game, we manually compute high granularity scores for the experiments in Tables 2, 3 in Appendix D.4. Our manual analysis, done by the authors, confirms that agents are not able to progress beyond the very beginning of each game (see Appendix Tables 10,11).

**Comparing results to other recent works.** Readers might wonder why Gemini 2.5 Pro struggles to complete even the first checkpoint in *Pokemon Crystal*, yet it was able to complete a very similar game, *Pokemon Blue*, in *Gemini Plays Pokemon* (Joel Z, 2025). This can be attributed to the stricter ruleset imposed in the VideoGameBench setting, as their agent scaffolding uses several special-built tools and overlays designed to assist the agent in 2D navigation and memory management. Furthermore, its scaffolding and prompting had been updated multiple times throughout the run, even sometimes being given direct instructions. Meanwhile, on VideoGameBench and VideoGameBench Lite, we explicitly disallow access to any information other than raw visual frames, including data sourced from the game state. We disallow placing visual overlays on the screen, such as maps of previously visited locations, and prohibit any form of human intervention while the agent plays the game. Nevertheless, we believe that game-specific scaffolds serve as a useful upper bound on game performance that can inform what capabilities future generalist agents require to solve our benchmark.

Table 3: Scores on the VideoGameBench Lite `test` split, a benchmark consisting of three games where the environment pauses while the agent is thinking. Each score is displayed as a percentage. 100% indicates a completed game.

| VideoGameBench Lite | GPT-4o | Sonnet 3.7 | Gemini 2.5 Pro | LLaMA 4 | Gemini 2.0 Flash | QwenVL-2.5 7B | QwenVL-2.5 32B |
|---|---|---|---|---|---|---|---|
| Doom II | 0% | 0% | 0% | 0% | 0% | 0% | 0% |
| Kirby's Dream Land | **4.8%** | **4.8%** | **4.8%** | 0% | 0% | 0% | 0% |
| Link's Awakening (DX) | 0% | 0% | 0% | 0% | 0% | 0% | 0% |
| Overall Score | **1.6%** | **1.6%** | **1.6%** | 0% | 0% | 0% | 0% |

**Ablating latency through VideoGameBench Lite.** In Table 3, we evaluate VG-Agent on the VideoGameBench Lite subset of games. Noticeably, every model performs considerably better both quantitatively and qualitatively on VideoGameBench Lite, even on games like *Doom II* where the models do not reach the first checkpoint (see Appendix D.4 for finer-grained quantitative scores). Nevertheless, the low overall performance suggests that even ignoring inference latency, VLM agents struggle to reason effectively over video game environments.

**Ablating game complexity with our practice game suite.** To investigate the large performance gap of VLM models on VideoGameBench, we devised three simple PC games. All games in VideoGameBench require visually identifying objects and their locations on screen, and many require clicking and dragging to perform tasks. Our *Location Clicking Game* is a game where players must click on 10 different markers that appear one at a time, at different locations on the screen. Finally, the *Mouse Dragging Game* consists of 10 different levels where the player must drag a marker along a path shown on-screen. The *2D Navigation game* is a grid-world setting where players use the arrow keys to move a marker through 10 different mazes. See Appendix F for screenshots of these games.

Table 4: Performance of frontier VLMs using the VG-Agent scaffold in three practice games we constructed, scored by how many targets (out of 10) each model could interact with in 250 actions. In the *Clicking Game*, the model must click 10 circles. In the *Dragging Game*, the model must drag 10 circles in a desired path. In the *2D Navigation Game*, the model must move a tile through 10 different grid mazes.

| Game | GPT-4o | Sonnet 3.7 | Gemini 2.5 Pro | LLaMA 4 | Gemini 2.0 Flash | QwenVL-2.5 7B | QwenVL-2.5 32B |
|---|---|---|---|---|---|---|---|
| Location Clicking | 30% | **100%** | **100%** | 10% | 0% | 0% | 10% |
| Mouse Dragging | 0% | **10%** | 0% | 0% | 0% | 0% | 0% |
| 2D Navigation | 30% | **80%** | 50% | 20% | 60% | 10% | 50% |

In Table 4, we find that Claude Sonnet 3.7 and Gemini 2.5 Pro are able to complete the *Location Clicking Game*, but struggle to complete the other games, while the other models struggle to complete all of the practice games. We tested our agent interface by having a co-author play the practice games through the human interface described in Appendix C.2. We were able to use this interface to complete all of our practice games, suggesting that the VLMs were not able to complete those games because of weaknesses in the VLMs and not because of weaknesses in our interface.

## 4.2 Qualitative Failure Modes on VideoGameBench

In this section, we address the low scores of frontier VLMs on VideoGameBench and VideoGameBench Lite by qualitatively analyzing game trajectories. We provide explicit action transitions in Appendix E for each of the examples we provide below. We hope these findings serve as potential future directions to explore for improving VLM capabilities on VideoGameBench.

**Solving VideoGameBench games requires narrowing the Knowing-doing Gap.** In *The Legend of Zelda: Link's Awakening*, all models observe that the door to leave the starter room is at the bottom of the screen, but will repeatedly press "down" irrespective of where the controlled character is in the room. Notably, in Paglieri et al. (2025), the authors observe a similar failure mode which they call the "knowing-doing gap", where a model knows what needs to be done at a given point but it carries out incorrect actions, underscoring a disconnect between knowledge and execution.

**Models frequently struggle to correctly process visual inputs.** In VideoGameBench, agents are only given access to the raw visual frames provided by the game emulator. We find that incorrectly processing the given input frames frequently leads to illogical behavior that can fatally compromise the run, such as the Gemini 2.0 Flash agent in *Doom II* wasting all its ammo on a dead enemy it misperceives as alive, or the GPT-4o agent in *The Legend of Zelda: Link's Awakening* mistakenly believing it had already spoken to an NPC because it had seen itself standing next to the NPC in a previous frame, despite no interaction (see Appendix E.2). These results align with prior studies on the shortcomings of VLM perception capabilities (Tong et al., 2024; Rahmanzadehgervi et al., 2025; Al-Tahan et al., 2024).

**VG-Agent lacks planning and memory management abilities.** We frequently observe frontier models struggling to track game state information and objectives, similar to findings in other visual game benchmarks like (Paglieri et al., 2025; Hu et al., 2025b;a). These limitations lead to incorrect decision making or repeated action loops so agents cannot progress further in the game – for example, in *Doom II*, VG-Agent with GPT-4o overwrites its textual scratchpad memory after reaching a new zone, deleting information on where its been before, and thus repeatedly traveling back and forth between the same zones. In *Pokemon Crystal*, VG-Agent with GPT-4o quickly forgets its primary objective to deliver a package and other information about its game state (see Appendix E.3).

## 5 Related Works

Game environments have long served as a testbed (Mnih et al., 2013) for deep learning agents, beginning with reinforcement learning approaches (Shao et al., 2019) and more recently extending to vision-language models that leverage multimodal inputs to interact with and reason about gameplay (Karten et al., 2025; Paglieri et al., 2025).

### 5.1 Reinforcement Learning to Play Games

Reinforcement Learning (RL) in has achieved significant success in game environments. Early work demonstrated RL agents capable of mastering Atari games from pixel inputs (Mnih et al., 2015). More advanced RL systems have achieved superhuman performance in complex board games like Go (Silver et al., 2016a) and real-time strategy games like StarCraft II (Vinyals et al., 2019). Even complex team-based games like Dota 2 have seen RL agents (OpenAI et al., 2019) beat professional players. These successes often rely on custom game APIs (Kempka et al., 2016; Vinyals et al., 2017), sophisticated reward shaping (Ma et al., 2024), simulated game trajectories (Ha & Schmidhuber, 2018; Hafner et al., 2024; Valevski et al., 2024), self-play (Silver et al., 2017), or learning from human data (Silver et al., 2016b; OpenAI et al., 2019; Vinyals et al., 2019). In 3D environments like Minecraft (Fan et al., 2022), model-based RL approaches such as Dreamer have shown promise in learning complex behaviors from visual inputs (Hafner et al., 2024). There have also been efforts to build generalist agents for interactive game environments through scale (DeepMind et al., 2024).

Recently, hybrid approaches combining LMs with RL techniques have emerged (Yao et al., 2020; NVIDIA et al., 2025), particularly for games with significant linguistic components. For instance, agents combining language models with RL policies have achieved human-competitive performance

in text-based battle simulators like Pokemon Showdown (Karten et al., 2025). However, traditional RL often struggles with extremely sparse rewards (Mnih et al., 2015), long-horizon tasks requiring complex reasoning or common sense (Küttler et al., 2020; Yao et al., 2020), and efficiently utilizing prior knowledge (Zhang et al., 2024), areas where foundation models might offer advantages.

## 5.2 VLMs and LMs as Game Playing Agents

VLMs and LMs offer a different paradigm for game playing, leveraging their pre-trained knowledge and reasoning capabilities to interpret visual scenes and text, formulate plans, and generate actions, often with minimal or no game-specific training (Tan et al., 2024; Ruoss et al., 2025). This reasoning paradigm is more akin to humans, which have been shown to heavily use priors for solving games (Dubey et al., 2018).

Early work explored LMs in text-based adventure games, demonstrating their potential for planning and interaction based on textual descriptions (Tuyls et al., 2022; Yao et al., 2022; 2023). Many benchmarks have also been developed to evaluate VLM agent capabilities such as visual-language understanding (Guan et al., 2024), 2D/3D motion planning (Gan et al., 2021; Nasir et al., 2024)

More recently, efforts have focused on applying VLMs to visually rich games. Some approaches simplify the environment, using object detectors or textual summaries (Zhu et al., 2023). Others attempt direct interaction with game interfaces, such as projects demonstrating models playing Pokemon based on screen captures (Rubinstein et al., 2025; Hershey, 2025). Agents have also been developed for games like Mario and Sokoban using VLM-based reasoning (Wu et al., 2023; Tan et al., 2024; Hu et al., 2025a). Another line of work includes LMs playing games that are traditionally hard for RL due to complexity or language elements (Paglieri et al., 2025).

Despite this progress, evaluating VLMs on complete, complex, real-time video games remains an open challenge. Existing benchmarks often focus on simplified environments Gan et al. (2021); Nasir et al. (2024), short tasks (Shridhar et al., 2021; Hu et al., 2025b), or lack the real-time interaction constraints of many popular games (Ruoss et al., 2025; Paglieri et al., 2025).

## 6 Discussion

**Limitations.** VideoGameBench covers primarily games and game emulators from the 1990s, which we hope to expand to more games and platforms. The automatic checkpoint detection we introduced works for measuring game progress, but is not able to provide extremely fine-grained or continuous markers. Lastly, we recognize that our wide range of mechanisms for preventing data leakage (e.g. a `dev` and `test` split of games, secret games, and strict scaffolding rules) could still be bypassed intentionally or unintentionally, since information, guides and video walkthroughs about virtually all video games are widely available online and might make its way to pre-training or finetuning corpora.

**Intellectual Property Usage.** The games in VideoGameBench fall under copyright laws, so care must be taken to ensure ownership. We cite prior academic works such as Mnih et al. (2013); Valevski et al. (2024); Hu et al. (2025a) that similarly introduce benchmarks or works around copyrighted games. For our checkpointing system, we use timestamps from available walkthroughs, but only use content from the original game frames, which falls under fair use.

**Broader Impact.** VideoGameBench evaluates whether AI systems can operate in new environments. Although we focus on the virtual game setting, improvements on VideoGameBench may translate into progress in real-world applications, including controlling autonomous robots and drones. We believe that evaluating and understanding agent capabilities in this virtual setting will be important for monitoring the potential for harmful real-world behavior in future systems.

**Conclusion.** Video games offer a uniquely rich and underexplored environment for evaluating model abilities: they require spatial reasoning, long-term memory, fast perception-action loops, and the ability to generalize across diverse scenarios. VideoGameBench leverages this setting to test whether vision-language models can exhibit the kinds of inductive biases humans use to complete novel tasks. We hope that this benchmark and future contributions will serve to build autonomous agents that are trustworthy, robust, and able to generalize to new tasks.

## ETHICS STATEMENT

As stated in the broader impact discussion, progress on VideoGameBench may translate into progress on real-world embodied AI such as robots and drones. We believe that evaluating agent capabilities in these settings will be important for understanding potential risks in future systems. We also address the usage of commercial video games and content in the main paper, but re-iterate that games in VideoGameBench fall under copyright laws. The use of YouTube walkthroughs for computing progress metrics in § 2.6 strictly uses video game frames and timestamps from the videos, and no other content from the video creators – all of this content falls under either fair use or copyright laws of the original games.

## REPRODUCIBILITY STATEMENT

The main experiments for VideoGameBench, VideoGameBench Lite, and the practice games described in § 4 are reproducible from our provided codebase. All experiments were run from model API providers using models with specified dates, and any reproducibility errors are due to either noise or changes to the API itself. Furthermore, we provide an anonymous link to the logs and actual runs used to generate all tables – all qualitative examples described in § 4.2 and the Appendix were taken from these logs. Finally, we include a script in the codebase to reproduce all main experiments in the tables. Due to issues described in the Ethics Statement above, we do not provide any game copies. We leave it to the user to follow copyright laws when running games on VideoGameBench, but we provide explicit instructions in our codebase for how to run experiments for these games when acquired.

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

# A  VIDEOGAMEBENCH RULES

In this section, we provide more details on the rules used for experiments run in Table 2 and Table 3. Firstly, we allow each agent to be prompted with basic information about the game setting and the game controls. Secondly, the game environment is only allowed to provide the raw game frames as input. Finally, we define a set of rules for determining when a run is over on VideoGameBench. We specifically allow runs to continue even after an agent has failed / "game over"d multiple times, and also note that some games do not have an end state other than completing the game (e.g. *The Incredible Machine*). We currently specify a simple rule of **20x** the time of the particular walkthrough for each game on VideoGameBench. For VideoGameBench Lite, we impose a similar restriction but assume **each step is one second of playtime**.

| VideoGameBench | Max Runtime |
| --- | --- |
| Zelda: Link's Awakening (DX) | 34:50:40 |
| Doom II | 81:44:00 |
| Kirby's Dream Land DX | 6:21:00 |
| Need for Speed | 30:32:20 |
| The Incredible Machine | 43:03:00 |
| Pokemon Crystal | 226:30:40 |
| Civilization | 4:58:12 |

| VideoGameBench Lite | Max Steps |
| --- | --- |
| Zelda: Link's Awakening (DX) | 101,440 |
| Doom II | 297,840 |
| Kirby's Dream Land (DX) | 21,603 |

Figure 3: To determine when a run ends in VideoGameBench, we provide a bound of $20\times$ the length of the scraped walkthroughs in Table 7. For VideoGameBench Lite, we use a similar metric, but determine time by translating one step to one second. Time is in the format (hours:minutes:seconds).

For information on the **stricter** set of rules we applied for our experiments due to budget constraints, see Appendix D.1.

# B  PROMPTS FOR VIDEOGAMEAGENT EXPERIMENTS

In this section, we provide all prompts used for our experiments. These prompts also serve as a starting point for future agents on VideoGameBench, and can be modified if they fall within the rules.

## B.1  EMULATOR-SPECIFIC PROMPTS

We provide prompts that are specific to a particular emulator. In the MS-DOS prompt for our experiments, we incorrectly included a paragraph about an overlay in our prompt, but note that this overlay **was not** used in any experiments in § 4.

**Instructions for MS-DOS Games**

```
You are a computer agent that uses the ReACT (Reasoning, Action,
Observation) framework with memory to play a video game.  For each
step, you should:
1.  Think:  Analyze the current state and decide what to do next
2.  Action:  Choose one of the following actions:
- click [options as action_input]:  Click the mouse at the current
mouse position.  Options include:
* right:  Right click instead of left click (default is left click)
* shift:  Hold shift while clicking
* ctrl:  Hold ctrl while clicking
* alt:  Hold alt while clicking
Multiple modifiers can be combined with +, e.g.  "shift+ctrl"
- move x,y:  Move the mouse to (x, y), where x is 0 on the left and
640 on the right, and y is 0 on the top and 400 on the bottom.
- drag x,y:  Drag (move while left button is down) to (x, y) from
the current mouse position, where x is 0 on the left and 640 on the
right, and y is 0 on the top and 400 on the bottom.
```

```
918
919    - scroll_up amount:  Scroll up by the specified amount
920    - scroll_down amount:  Scroll down by the specified amount
921    - write text:  Type the specified text
       - press_key key:  Press a specific key or key combination
922    - hold_key key[,duration]:  Hold a key down for a specific duration
923    (default 0.5 seconds)
924    3.  Observation:  You will receive the result of your action
925
926    You will interact with the game via the keyboard and mouse actions.
       To help you with mouse actions, we provide a thin red grid overlay
927    that intersects the screen at 100x100 pixel intervals (labelled
928    with coordinates divided by 100).  I also added 4 blue dots 25
929    pixels away in each direction with their exact coordinates in
       case you get lost.  The coordinates start at (0,0) at the top
930    left of the screen, indexed (x,y) and go up to (640,400) at the
931    bottom right.  For example, if you want to click somewhere inside
932    a box with top left corner at (100,100) and bottom right corner at
933    (150,150), you can move to (125,125) then click (estimate based on
934    the picture!  Try to get it as close as possible, don't rely on
       multiples of 10).
935
936    For keyboard actions, use the following format:
937    - Single keys:  "KeyA", "KeyB", "Digit1", "ArrowLeft", "ArrowUp",
938    "Enter", "Escape", "Backspace", "Tab", "Space"
       - Special keys:  "Control", "Alt", "Shift", "Meta"
939    - Key combinations (use + symbol):  "Control+KeyC", "Shift+ArrowUp",
940    "Alt+F4"
941    - Sets of Key combinations (multiple keys pressed at the same time):
942    "KeyA,Shift+KeyB"
943
944    Respond in the following JSON format:
945    {
946      "thought": "your reasoning about what to do next",
947      "action": "one of the available actions",
       "action_input": "parameters for the action",
948      "memory": "important information to remember for future steps"
949    }
950    To not update memory, respond with an empty string.
951
952    For example:
953    {
954      "thought": "I need to left click on the search box",
955      "action": "click",
       "action_input": "",
956      "memory": "1. My short term plan is to capture the enemy flag.\n
957    2. My opponent is trying to block my path, I should be wary.\n
958    3. Farms make my units stronger.
       4. The M button is to move units."
959    }
960
961    Another example of right clicking:
962    {
963      "thought": "I need to right click on the search box",
       "action": "click",
964      "action_input": "right",
965      "memory": ""
966    }
967    Or for keyboard actions:
968    {
969      "thought": "I need to move the character left in the game",
970      "action": "press_key",
971      "action_input": "ArrowLeft",
```

```
    "memory": "The character moves faster when holding the arrow key
    down instead of tapping it."
}
Do NOT wrap anything in "`json"` tags, and only respond with the
JSON object.
Always analyze the screenshot carefully to determine the correct
coordinates for your actions.
The memory field should contain any important information you want
to remember for future steps.
```

We also separately prompt the Game Boy emulator games for the Lite version of the benchmark. Although we could have done the same for the MS-DOS games, we chose not to.

**Instructions for Game Boy Games**

```
You are an AI agent playing a Game Boy game.  You will receive game
screens as images and must decide what buttons to press.  Feel
free to skip the start screen and play a new game.  Your goal is
to play the game effectively by analyzing the visual information
and making appropriate decisions.  You should respond with a list
of (or single) actions to perform in sequence (each is performed
for roughly 1/4 second) ard wrapped in "`actions"` tags.  You
can repeat actions to maintain pressure on the buttons.  To press
multiple buttons simultaneously, group them in a tuple.
Example response format:  "`actions

[
    "A",             # Press A button
    ("B", "UP"),     # Press B and UP simultaneously
    "RIGHT",         # Press RIGHT
    "START",         # Press START
    ("A", "B", "DOWN")  # Press A, B, and DOWN simultaneously
]
```

**Instructions for Game Boy Games on VideoGameBench Lite**

```
You are an AI agent playing a Game Boy game.  You will receive game
screens as images and must decide what buttons to press.  Feel free
to skip the start screen and play a new game.  Your goal is to
play the game effectively by analyzing the visual information and
making appropriate decisions.  You should respond with a list with
a single (or tuple of) buttons to press for the Game Boy emulator
(each is performed for roughly 1/2 second or 30 frames) ard wrapped
in "`actions"` tags.  Please do not add comments to the response.
Example response format (press A twice):
```actions
[
    ("A"),
    ("A"),
]
```

Another example of pressing multiple buttons simultaneously:

```actions
[
    ("A", "B", "DOWN")
]
```

Never press START and SELECT simultaneously, as this will restart
the emulator.
Available buttons:  A, B, START, SELECT, UP, DOWN, LEFT, RIGHT
```

## B.2 GAME-SPECIFIC PROMPTS

We include game prompts provided to VG-Agent while playing each game on VideoGameBench (specifically the `test` split used in § 4.

---

**Instructions for "Civilization I"**

```
You are playing Civilization on DOS. Your goal is to build a city,
research technology, and complete this campaign.
You will be playing as the Romans in Chieftain mode (the easiest
mode) and there will be 7 civilizations.

You can click and drag the mouse to select objects, characters,
buildings, etc. and also use your keyboard keys to move.
Game Objectives

Win Conditions: Achieve global dominance by conquering all other
civilizations or be the first to send a spaceship to Alpha Centauri.

Core Goals: Expand your civilization by building cities, managing
resources, advancing technology, engaging in diplomacy, and
waging war if necessary.

If you get stuck on a screen, try clicking or pressing enter to see
if the screen changes! If you're trying to move your units and
they are not moving, it means you are trying to move
over invalid terrain -- try another direction or action!

Make sure to remember all of the following facts:
1. Ground troops cannot walk through water (the blue regions),
mountains, or other obstacles.
2. End your turn when you're finished with what you want to do.
3. Each unit moves 1 tile. So if you want to move another unit,
move the selected unit first.
4. In the beginning, a good strategy is to just explore and have
your units move around and explore unseen areas.

General Controls
    Mouse: Click to select units, cities, and menu options.
    Right-click may provide additional info.

    Keyboard Shortcuts:
        Movement: Arrow keys (or Numpad) to move selected unit.
        End Turn: Enter key.
        Access City Menu: Click on a city or press C.
        Change Tax/Science/Luxury Rates: F1 (Tax Advisor).
        Access Civilopedia: F10.
        Save Game: Shift + S.
        Load Game: Shift + L.
        View Military Advisor: F3.
        View Diplomacy Screen: F4.
        View World Map: F5.

City Management
    Found City: Move settler unit to an empty tile and press B.
    Manage Production: Click on a city, then choose what to build
    (units, buildings, wonders).
    Adjust Citizen Tasks: Click on tiles within city radius to assign
    workers.

Unit Management
    Move Units: Use arrow keys or Numpad.
    Fortify Unit: Press F to make the unit stay in place and defend.
```

```
    Skip Turn: Spacebar.
    Disband Unit: Press D.
    Activate Next Unit: N key.

Combat & Diplomacy
    Attack Enemy: Move military unit onto an enemy unit or city.
    Negotiate Diplomacy: Open diplomacy screen (F4) and
    select a civilization to negotiate with.
    Declare War: Refuse enemy demands or attack their units/cities.

Research & Progression
    Select New Technology: Open the Science Advisor menu (F6)
    and choose a research path.
    Advance Eras: Progress by researching key technologies.
    Build Wonders: Unique, powerful structures that provide
    long-term benefits.

Unit Management
    Move: Arrow keys or Numpad (1-9, except 5) for movement.
    Fortify (F): Increases defense by 50%, unit remains stationary.
    Sentry (S): Unit does not require orders each turn but activates
    when an enemy approaches.
    Skip Turn (W): Temporarily skip a unit and return to it later.
    No Movement (Spacebar): Ends the unit's turn without action.
    Activate Fortified/Sentry Unit: Click the unit or
    select from the city menu.
    Unload from Ship (U): Unloads units from a transport ship.
    Go To Command (G): Orders the unit to move to a specified
    location.
    Change Home City (H): Assigns a new home city,
    used for air unit refueling.
    Disband (Shift + D): Removes the unit permanently.
    Pillage (Shift + P): Destroys improvements on a tile.

City Management
    Found a City (B): Settler establishes a new city.
    Grow a City (B in existing city): Settler increases city
    population.
    Manage Production: Select what to build (units, buildings,
    wonders).
    Adjust Citizen Tasks: Assign workers to different tiles.
    Set Tax/Science/Luxury Rates (F1): Adjust economy.
```

## Instructions for "The Need for Speed"

```
You are playing Need for Speed. You are sitting in your car
at the starting line, ready to compete.
Your goal is to win the race on each track.
You need to shift gears (up) when starting the race to move forward.
Move around the menu screen using the arrow keys (up and down),
press Enter to start.

Controls:
- Steering: Arrow keys (← →) or move mouse left/right
- Accelerate/Brake: Up/Down arrows (↑ ↓) or Left-click/Right-click
mouse
- Shift Gears: A (shift up) / Z (shift down)
- Hand Brake: Spacebar
- Horn: H
- Camera Views: C
- Pause: ESC or P
```

```
Additional Function Keys:
- F1: Toggle window size
- F3: Toggle view distance
- F5: Mute sound
- F7: Toggle status bar
- F9: Toggle dashboard

To go forward after starting the race, you need to first shift
gears up (e.g. press A), then hold up arrow,
e.g. hold_key ArrowUp 5000 (or some other large number).
Be careful not to select too high of a number
so you can react and turn (you need to move forward while turning
to drift).

Get ready to race! Use your skills to outmaneuver opponents
and claim victory on every track.
```

## Instructions for "The Incredible Machine"

```
You are playing The Incredible Machine, a puzzle game where you must
use various mechanical objects and devices to solve Rube
Goldberg-style challenges.
Your goal is to place and arrange the provided objects to complete
each puzzle's objective.

Game Controls:
Mouse Controls:
    Left Click: Select and place objects
    Right Click: Remove placed objects
    Click and Drag: Move objects around the puzzle area

Keyboard Controls:
    Space: Start/Stop the machine
    R: Reset the puzzle
    ESC: Access menu
    F1: Help screen

Game Mechanics:
- Each puzzle provides specific objects you can use
- Objects must be placed in the correct positions to create chain
reactions
- Physics affects how objects interact (gravity, momentum, bouncing)
- Some objects need to be precisely positioned to work properly

Common Objects:
- Ropes and pulleys: Transfer motion
- Conveyor belts: Move objects horizontally
- Springs: Bounce objects
- Motors: Provide continuous rotation
- Gears: Transfer rotational motion
- Balls and bowling balls: Roll and bounce
- Cats and mice: Animals that react to each other
- Balloons: Float upward
- Electrical switches: Trigger connected devices

Problem-Solving Tips:
1. Examine the puzzle goal carefully
2. Study the available objects
3. Consider how objects will interact
4. Test your solution in parts
5. Make small adjustments for timing
6. Watch for unintended interactions
```

```
7. Use gravity to your advantage

Remember:
- There are often multiple solutions to each puzzle
- Timing is crucial for many puzzles
- Some objects may not be needed
- Pay attention to object orientation
- Chain reactions should flow naturally
- Save working parts while experimenting with others

If stuck:
- Reset and try a different approach
- Watch how objects interact during test runs
- Break down complex solutions into smaller steps
- Consider unconventional uses for objects
- Look for visual hints in the puzzle design
```

## Instructions for "Pokemon Crystal"

```
You are playing Pokemon Crystal version. Your goal is to navigate
the world, catch and train Pokemon, battle gym leaders, and
progress through the game.
You start as a young trainer in Pallet Town, choosing between
Bulbasaur, Charmander, or Squirtle as your starter Pokemon.
Your ultimate goal is to defeat the Elite Four and become the
Pokemon Champion, while completing your Pokedex by catching
all 151 original Pokemon.

Analyze the current game screen and decide what buttons to press.
Respond with a sequence of actions to perform.
Think step by step:
1. What is happening in the current screen?
2. What action would be most appropriate?
3. What buttons need to be pressed to take that action?

Available buttons: A, B, START, SELECT, UP, DOWN, LEFT, RIGHT

Tips:
1. To get past any menu or typing screen, press START or START, A
when you are done. No matter where your arrow is on the screen,
it'll go to the end.
2. When trainers see you, they will want to battle.
3. In a Pokemon battle, you attack your enemies and you lose if your
Pokemon all reach 0 HP.
4. When typing a name, just press A twice to exit when your name
is full.
Don't go right then A.
5. Wild Pokemon appear randomly when walking in tall grass, caves, or
while surfing.
6. During battles (using the movement keys to move icons):
    - Choose "FIGHT" to use your Pokemon's moves
    - Choose "BAG" to use items like Potions or Pokeballs
    - Choose "POKEMON" to switch to a different Pokemon
    - Choose "RUN" to attempt escaping from wild Pokemon battles
7. Type advantages are crucial: Water beats Fire, Fire beats Grass,
Grass beats Water
8. Use Pokemon Centers (buildings with red roofs) to heal your
Pokemon for free
9. Buy supplies like Pokeballs and Potions at PokeMarts
(buildings with blue roofs)
10. Read dialogue and continue by pressing 'A'.
```

Each movement key (e.g. UP, DOWN, LEFT, RIGHT) will move your
character (with the hat) one tile in that direction.
Keep that in mind, and calculate where to go based on what
you want to do.

You can interact with people (you should to get information and also
proceed in the game) using the A button by standing next to them.

## Instructions for "Doom II"

You are playing Doom II on DOS. Your goal is to complete the game by
defeating all the levels. Explore all rooms as much as possible.
The enemies are wearing red and green with dark green heads. If
they are not moving, they are probably dead. Do not confuse
them with random objects.

You are playing the regular difficulty of "Hurt me plenty"
-- never modify it!

Do NOT FORGET the following:
1. Look for doors, which will be in the corridors and have some
kind of writing on the door (e.g. UAC is a door). You can
open them!
Try aligning yourself so the door is centered on your screen,
then walk up to it.
When you're pressed against the door, press space to open it.
2. Doors usually have blue triangles (they themselves are not doors)
near them on the sides, and it will be obvious you can open it.
3. You need to be directly in front of the door and press 'space' to
open it, you cannot be far away to open it. Don't just go
backwards because you're not sure.
4. If you get stuck on a wall or moving against a wall, try taking
a few steps back and re-adjusting your thoughts.
5. If there are a lot of enemies or you are being shot at, try
strafing around and moving a lot side to side to avoid
getting fired at while also aiming and shooting.

Remember exactly what direction you were turning so you don't make
redundant movements. Use the repeated key presses to turn.
You can also move your character to adjust your aim. If YOU SHOT
AND IT DID NOTHING, IT PROBABLY MEANS YOU WERE OFF TARGET. Re-adjust.

Keep moving forward until your screen doesn't change.
Do not gaslight yourself into thinking you're stuck at a door when
you are not. You will know easily when you are stuck.

You should aim and kill any enemy you see, do not just walk past them,
as they will shoot you once you pass them. If it does walk
past you, remember that it is now behind you in your memory -- do
not forget, and turn around and make sure you eliminate them.

If your screen flashes red it might mean that you're being shot,
turn around if you dont see an enemy in front of
you to check is
an enemy is shooting you from the back.

Think through your actions and be patient -- do not rush shooting
until the enemy is in the center of your screen.

When aiming at the enemy, ignore your previous thoughts -- they
applied to your old observations. For example, if they
were centered in your thoughts before,

```
they might not be centered anymore, and you will need to re-adjust.
Look at your most recent observation to make sure.
Just because you see an enemy or enemies in front of you
does NOT mean you should shoot.
Carefully align your gun so they are exactly at the center.
Don't just blindly shoot.
Focus on ONE enemy at a time. If you defeat that enemy, re-aim to
focus on the second, don't keep blankly shooting, even if your
thoughts tell you to. You need to aim.

Move with the arrow keys to adjust your aim, more controls below.
In general, you can chain multiple keys in sequence (e.g. if the
enemy is slightly to the right, ArrowRight,Control)
Basic Controls:
- Repeat `ArrowLeft` or `ArrowRight` 14 times separated by commas
to turn left and right 90 degrees if you want to turn a corner.
- Repeat W multiple times separated by commas,
e.g. press_key W,W,W,W,W,W if you want to explore forward.
- Control: Fire weapon
- Space: Open door / interact

For finer control when dealing with enemies:
- ArrowLeft, ArrowRight: Look left and right to adjust your aim and
look around
- W, A, S, D: Move forward, left, back, right
- Shift + W, A, S, D: Run forward, left, back, right
```

### Instructions for "Kirby's Dream Land"

```
You are playing Kirby's Dream Land on the Game Boy. You control
Kirby, a round hero who can walk, jump, inhale enemies, and fly.
Your goal is to progress through levels, defeat enemies, and
overcome bosses to eventually save Dream Land.

Analyze the current game screen and decide what buttons to press.
Think through each situation step by step:

1. Assess the current screen:
    - What enemies or obstacles are present?
    - Is Kirby on the ground or in the air?
    - Are there any platforms or doorways?
    - Is there a boss battle happening?

2. Consider your options:
    - Do you need to avoid enemies?
    - Should you inhale enemies to use as projectiles?
    - Is flying a better option than walking?
    - Are there items or power-ups to collect?

3. Plan your next action and execute using the available controls:

MOVEMENT CONTROLS:
- LEFT/RIGHT on Control Pad: Move Kirby left/right
- UP on Control Pad: Enter doorways or fly upward
- DOWN on Control Pad: Crouch and swallow inhaled enemies

ACTION BUTTONS:
- A Button: Jump
- B Button: Inhale enemies/objects or spit them out as projectiles
- START Button: Pause the game
```

```
VITALITY AND CHANCES:
- Kirby has 6 vitality bars that decrease when hit by enemies
- Losing all vitality bars costs one chance (life)
- Game ends when all chances are depleted

SPECIAL NOTES:
- You can float indefinitely by repeatedly pressing the A button
- Inhaled enemies can be used as projectile weapons
- During boss battles, watch the boss's vitality bar above Kirby's
- If stuck, you can reset the game by pressing A, B, START, and
SELECT simultaneously

A few things you should ALWAYS remember:
1. For things that say "IN" or black doors / light doors,
Kirby has to go into it to go into a room.
Don't just hover above it.
2. Shining stars (called warp stars) are the end of the level, and
transition you further into the game. Kirby has to go into it or
step on it.
3. Do not hit enemies directly, or Kirby will take damage.
Spit out enemies (not bosses) or items like
bombs back to damage your enemies!

Kirby is a classic platformer, so you generally should continue
to the right to progress in the game.
Respond with a clear sequence of actions, explaining your reasoning
for each decision.

Available buttons: A, B, START, SELECT, UP, DOWN, LEFT, RIGHT
```

**Instructions for "The Legend of Zelda: Link's Awakening (DX)"**

```
You are an expert playing The Legend of Zelda: Link's Awakening
on Game Boy. Your goal is beat the game flawlessly by navigating
the world, solving puzzles, defeating enemies, and progressing
through dungeons. You cannot walk through walls or doors,
so try stepping back or around them!

Controls:
- Arrow keys: Move Link around, move options around in a menu.
- A: Use Item Box A. Also for proceeding with dialogue.
- B: Use Item Box B.
- START: Open inventory.
- SELECT: View map or switch items

Analyze the current game screen and decide what buttons to press.
Respond with a sequence of actions to perform.
Think step by step:
1. What is happening in the current screen?
2. Are there enemies, NPCs, or interactive objects?
3. What action would help progress in the game?
4. What buttons need to be pressed to take that action?

You cannot move if dialogue is on the screen until you finish it,
so keep pressing A until it is over.

Available buttons: A, B, START, SELECT, UP, DOWN, LEFT, RIGHT
```

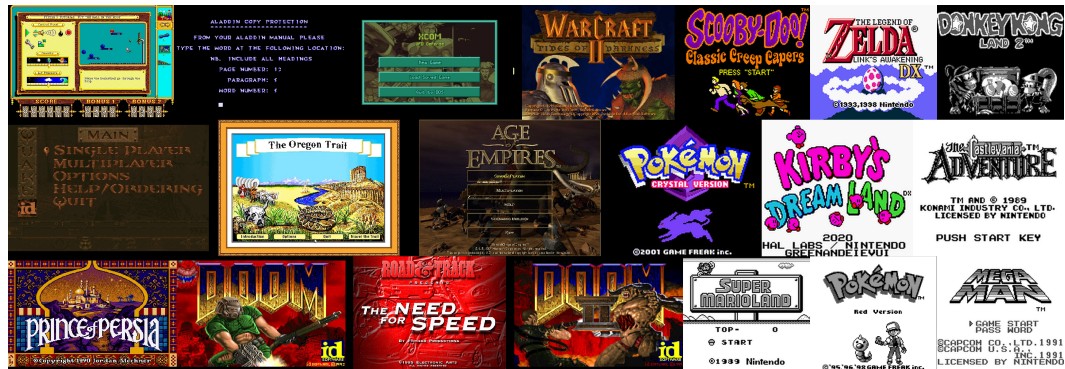

Figure 4: VideoGameBench features a set of 20 video games from the MS-DOS and Game Boy platforms that VLMs are tasked with playing to completion.

## C  VIDEOGAMEBENCH DETAILS

In this section, we provide additional information on the VideoGameBench setup. We show all the public video games in the `dev` and `test` splits of VideoGameBench in Figure 4. We also provide more information of checkpoint detection.

### C.1  LIST OF AVAILABLE ACTIONS

Here, we list all the actions available that an agent can take on VideoGameBench for both Game Boy and MS-DOS games. For key or button presses, we explicitly allow the agent to specify how long they hold the keys for, but found this to be confusing because it is difficult to embed time into the agent's context. We use a default time of half a second for key / button presses, but provide the option to change this default setting in the benchmark. We also allow agents to specify the length of their key presses.

#### C.1.1  LIST OF GAME BOY ACTIONS

Using the PyBoy (Mads Ynddal, 2025) interface, we enable agents on VideoGameBench to press the following buttons:

- `A` — Press the **A** button on the Game Boy emulator
- `B` — Press the **B** button on the Game Boy emulator
- `SELECT` — Press the **SELECT** button on the Game Boy emulator
- `START` — Press the **START** button on the Game Boy emulator
- `RIGHT` — Press the right arrow on the Game Boy emulator D-Pad
- `LEFT` — Press the left arrow on the Game Boy emulator D-Pad
- `UP` — Press the up arrow on the Game Boy emulator D-Pad
- `DOWN` — Press the down arrow on the Game Boy emulator D-Pad

#### C.1.2  LIST OF KEYBOARD AND MOUSE ACTIONS

Using the Playwright (Mads Ynddal, 2025) interface with JS-DOS (Guryanov, 2025), we enable agents to use the mouse and keyboard interface in the following way:

- `click` — Click the mouse at the current position. You can specify options like right-click or modifier keys (e.g., Shift, Ctrl, Alt).
- `move` — Move the mouse to a specific 2D pixel coordinate on the screen, e.g., `"150,200"`.
- `drag` — Drag the mouse to a target coordinate, simulating a click-and-drag motion.

- `scroll_down` — Scroll down by a specified number of pixels.

- `scroll_up` — Scroll up by a specified number of pixels.

- `write` — Type a string of text as keyboard input into the emulator.

- `press_key` — Press a single key or a sequence of keys (e.g., `"Enter"`, `"Ctrl,C"`).

- `hold_key` — Press and hold a key for a specified duration in seconds (e.g., `"A,1.5"` to hold `A` for 1.5 seconds).

### C.2 VERIFYING THAT VIDEOGAMEBENCH IS CAPABLE OF BEING COMPLETED

While each game in VideoGameBench has been completed by at least one human, we also provide evidence that our interface does not cause games to be impossible to solve. Using *PyBoy* and *Playwright*, we give agents sufficient access to press any button or set of buttons simultaneously that the emulator allows. Our human interface therefore enables the same controls as the emulator.

We focus instead on the information given to the agent, and only provide the human with the same information. In other words, the human only sees the most recent frame before taking an action. To simulate what a model "sees" before making an action, we only show the most recent frame to the human, who must make an action based on this frame and their own memory. Furthermore, the interface waits at least half a second before the action is made and the new frame is updated to the player. The emulator runs entirely in real-time, but the human is limited both in controls and perception by this fixed delay. Using this interface, the authors manually beat the first checkpoint of *Kirby's Dream Land* and *Doom II*, both of which require real-time inputs.

### C.3 SCRAPING CHECKPOINTS FROM WALKTHROUGHS

We provide automatically trackable progress markers on VideoGameBench by scraping checkpoint frames from online available walkthroughs. In previous VLM-based playthroughs of video games such as *Gemini Plays Pokemon* (Joel Z, 2025), progress was manually marked by viewers. To help scale these experiments to several different games, we provide a system of finer, automatic tracking of progress.

Each video game in VideoGameBench has an associated walkthrough available through video streaming platforms such as YouTube. These playthroughs contain full, unedited gameplay of a human playing each video game to completion. Many of these videos contain timestamps with game checkpoints that we used to determine checkpoints, but we also scrape other suitable checkpoints as well. To add a checkpoint to a game, we require a (`image-frame`, `timestamp`) pair, which can be found directly from these videos.

### C.4 HOW CHECKPOINTS WERE DETERMINED

All games in VideoGameBench have a linear progression, which means there is a certain pre-defined path that a player must follow to complete the game. For example, while there are infinite possible trajectories in the game *Pokemon Red*, there are certain unavoidable story checkpoints (e.g. grab each Gym badge, defeat the Elite 4, etc.) that the player must reach. Furthermore, many of these story checkpoints are actually rendered the same. For example, when a player in *Pokemon Red* defeats a Gym Leader, the game will always render the same background with the player facing the Gym Leader and the same text, regardless of the previous trajectory taken by the player up to this point. We use these frames as checkpoint frames, and estimate progression based on when an online walkthrough has reached this frame.

In our setting, the easiest checkpoints to scrape are typically time-stamped for online walkthroughs, but there are often more available checkpoint frames that can be used. Adding more checkpoint frames increases the granularity of automatic progress detection, which we plan to expand on in the future.

Table 5: List of all checkpoints and their corresponding scores (as a percentage) on VideoGameBench. The only exception is The Incredible Machine, which scores each level equally from Levels 1–87.

| **Test Set Game** | **Checkpoint** | **Score (%)** |
|---|---|---|
| Kirby's Dream Land DX | Reach first warp star / mini-boss | 4.81 |
| | Reach Stage 2 - Castle Lololo | 13.82 |
| | Reach Lololo Mini-Boss | 20.21 |
| | Reach Stage 3 - Float Islands | 28.17 |
| | Reach Stage 4 - Bubbly Clouds | 43.48 |
| | Reach Kracko Jr. Mini-Boss | 51.62 |
| | Reach Stage 5 - Mt. Dedede | 66.58 |
| | Finish Credits | 100.00 |
| Pokémon Crystal | Obtain first Pokemon from Oak | 0.93 |
| | Defeat Johto Gym #1 | 8.85 |
| | Defeat Johto Gym #2 | 14.22 |
| | Defeat Johto Gym #3 | 21.85 |
| | Defeat Johto Gym #4 | 28.84 |
| | Defeat Johto Gym #5 | 34.75 |
| | Defeat Johto Gym #6 | 36.82 |
| | Defeat Rocket Executive | 41.17 |
| | Defeat Johto Gym #7 | 43.40 |
| | Disband Team Rocket | 50.95 |
| | Defeat Johto Gym #8 | 57.76 |
| | Kanto Champion (Defeat Lance) | 67.23 |
| | Defeat Kanto Gym #1 | 70.69 |
| | Defeat Kanto Gym #2 | 73.52 |
| | Defeat Kanto Gym #3 | 80.74 |
| | Defeat Kanto Gym #4 | 84.20 |
| | Defeat Kanto Gym #5 | 86.73 |
| | Defeat Kanto Gym #6 | 89.75 |
| | Defeat Kanto Gym #7 | 93.52 |
| | Defeat Kanto Gym #8 | 95.22 |
| | Defeat Red | 100.00 |
| Legend of Zelda DX | Grab Sword from the Beach | 1.08 |
| | Collect the Full Moon Cello | 10.32 |
| | Collect the Conch Horn | 16.68 |
| | Collect the Sea Lily's Bell | 31.12 |
| | Collect the Surf Harp | 48.76 |
| | Collect the Wind Marimba | 62.85 |
| | Collect the Coral Triangle | 76.04 |
| | Collect the Organ of Evening Calm | 86.42 |
| | Collect the Thunder Drum | 96.17 |
| | Complete the Game (reach Owl) | 100.00 |
| Civilization I | Win a game on Chieftain, 7 civilizations | 100.00 |
| Need for Speed | Get 1st on "City" | 12.54 |
| | Get 1st on "Coastal" | 24.19 |
| | Get 1st on "Alpine" | 37.81 |
| | Get 1st on "Rusty Springs" | 44.88 |
| | Get 1st on "Autumn Valley" | 54.68 |
| | Get 1st on "Burnt Sienna" | 65.92 |
| | Get 1st on "Vertigo Ridge" | 77.57 |
| | Get 1st on "Transtropolis" | 89.54 |
| | Get 1st on "Lost Vegas" | 100.00 |
| The Incredible Machine* | Solve Level 1 | 1.12 |
| | Solve Level 2 | 2.24 |
| | Solve Level 3 | 3.37 |
| | : | : |
| | Solve Level 87 | 100.00 |

Table 6: Continuing from Table 5, Doom II checkpoints and their corresponding scores (as a percentage) on VideoGameBench.

| **Test Set Game** | **Checkpoint** | **Score (%)** |
|---|---|---|
| | Beat Stage 1 | 0.81 |
| | Beat Stage 2 | 2.31 |
| | Beat Stage 3 | 3.91 |
| | Beat Stage 4 | 5.25 |
| | Beat Stage 5 | 8.04 |
| | Beat Stage 6 | 11.21 |
| | Beat Stage 7 | 12.24 |
| | Beat Stage 8 | 15.57 |
| | Beat Stage 9 | 20.10 |
| | Beat Stage 10 | 21.63 |
| | Beat Stage 11 | 28.84 |
| | Beat Stage 12 | 32.33 |
| | Beat Stage 13 | 37.16 |
| | Beat Stage 14 | 39.75 |
| Doom II | Beat Stage 15 | 48.25 |
| | Beat Stage 16 | 51.41 |
| | Beat Stage 17 | 55.98 |
| | Beat Stage 18 | 59.83 |
| | Beat Stage 19 | 63.73 |
| | Beat Stage 20 | 66.00 |
| | Beat Stage 21 | 69.39 |
| | Beat Stage 22 | 71.20 |
| | Beat Stage 23 | 74.33 |
| | Beat Stage 24 | 80.21 |
| | Beat Stage 25 | 82.64 |
| | Beat Stage 26 | 85.66 |
| | Beat Stage 27 | 90.99 |
| | Beat Stage 28 | 94.60 |
| | Beat Stage 29 | 98.52 |
| | Beat Stage 30 | 100.00 |

## C.5 PERCEPTUAL IMAGE HASHING FOR CHECKPOINT DETECTION

We provide a baseline hamming distance (Hamming, 1950) threshold of $< 12$ for determining if two frames match, but enable the option to tune this parameter individually for each checkpoint in each game. For example, for *Kirby's Dream Land*, we choose between $6 - 8$ for this threshold.

One limitation of applying perceptual hashing over full game screen images is when distinctions between checkpoints are local to a certain patch of the image. For example, in *The Incredible Machine*, the level completion screen is a tiny textbook that indicates that a level was completed. We are planning to implement features for defining a rectangular crop of the screen and comparing hashes for these crops rather than the full screen.

Another limitation that requires tuning is that scraped frames may have slight differences or artifacts than the frame in the playthrough. For example, the frame may contain the current player's health bar such as in *Kirby's Dream Land*, which may differ from the health bar in thte walkthrough at that frame. Other issues include different aspect resolutions from scraped walkthroughs, all of which can be solved with manual tuning. We want to provide a robust system for automatic checkpoint detection as scores on VideoGameBench continue to improve.

## C.6 BENCHMARK STATISTICS

We provide a distribution for the length of each game in VideoGameBench, as well as the checkpoints scraped for the test split. In Figure 5, we show the distribution of checkpoints on VideoGameBench.

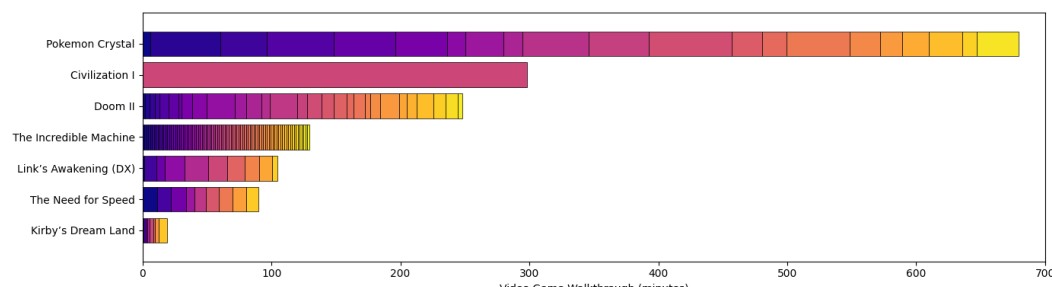

Figure 5: **VideoGameBench checkpoint lengths.** We show the length of each game walkthrough and the position of each checkpoint as a black divider. Checkpoints in VideoGameBench are mapped to the timestamp it was scraped from in an online walkthrough video to determine the percentage of the game that was completed.

### C.7 LINKS TO WALKTHROUGHS

We provide links in Table 7 to all walkthroughs used for scraping checkpoints, as well as estimating game lengths. Each of these walkthroughs assumes full knowledge of the game and does not consider time spent exploring.

Table 7: List of longplay walkthrough links for scraping checkpoints on VideoGameBench.

| **Test Set Games** | **Walkthrough Link** |
|---|---|
| Legend of Zelda DX | https://www.youtube.com/watch?v=rLFXGs1Rr6c |
| Doom II | https://www.youtube.com/watch?v=nhmRxFf02JA |
| Kirby's Dream Land DX | https://www.youtube.com/watch?v=n8CSolb0hjc |
| Civilization I | https://www.youtube.com/watch?v=XnfkrhREDDM |
| Need for Speed | https://www.youtube.com/watch?v=1Sf7_TbG8Js |
| The Incredible Machine | https://www.youtube.com/watch?v=pTbSMKGQ_rU |
| Pokémon Crystal | https://www.youtube.com/watch?v=HQEaaIuyKAM |
| **Dev Set Games** | |
| Doom | https://www.youtube.com/watch?v=Q4GiCg_m8wA |
| Quake | https://www.youtube.com/watch?v=Y8k9c-6Me_A |
| Civilization | https://www.youtube.com/watch?v=o6_5PIsJkYk |
| Warcraft II | https://www.youtube.com/watch?v=DWjZQzviTUs |
| The Oregon Trail Deluxe | https://www.youtube.com/watch?v=FaWmldKoo9Y |
| X-COM UFO Defense | https://shorturl.at/BPIEb |
| Scooby Doo: Classic Creep Capers | https://www.youtube.com/watch?v=yYfpaz9NwU8 |
| Prince of Persia | https://www.youtube.com/watch?v=qMOBiT3F6AM |
| Age of Empires | https://www.youtube.com/watch?v=hqsbQzpFRBI |
| Pokémon Red | https://www.youtube.com/watch?v=hYcotDHI0dg |
| Super Mario Land | https://www.youtube.com/watch?v=vfaV2VDBQEk |
| Castlevania: The Adventure | https://www.youtube.com/watch?v=oL5TwYG0iQ8 |
| Donkey Kong Land 2 | https://www.youtube.com/watch?v=tqUU9pwUxQo |
| Mega Man: Dr. Wily's Revenge | https://www.youtube.com/watch?v=wpwESPNjAk0 |

## D EXPERIMENT DETAILS

In this section, we provide additional information on the experiments in § 4. We provide criterion we used to determine the end of a run, cost information for the main experiments, and a variance analysis of for the best performing model.

### D.1 CRITERION FOR ENDING RUNS

Due to budget constraints, the experiments in § 4 were ended when the agent was stuck or not progressing for a certain period of time. In this section, we detail exactly the constraints we used per game to determine whether a run should be pre-maturely ended. We ended a run if:

1. The agent **quits the emulator or puts the game in a "locked" state**. In DOS or GBA games, if the agent quits the entire game (which requires multiple steps), it is unable to restart the game.

2. The game provides the agent multiple lives, and the agent has a "Game Over" screen by losing them all (e.g. *Kirby's Dream Land* and *Super Mario Land*). Otherwise, if the agent loses in the same location more than three times, we also end the run.

3. The agent is "stuck" (i.e. the exact same screen) for more than 100 steps. We reason that the context window is only 20 steps, so the conditional probability on this context of getting unstuck is low.

4. The agent loses without damaging any enemies, indicating little progress can be made in multiple repeated trials.

5. The agent uses $30 ($\sim$ 2000 steps) without reaching a new checkpoint. This was mainly to avoid long loops like in *Doom II* where the agent repeatedly revisits the same locations.

### D.2 MAIN VIDEOGAMEBENCH EXPERIMENTS COST

We report the cost per experiment of Table 2. Discrepancies in costs do not necessarily reflect differences in model costs – rather, some runs were ended early according to the criteria in Appendix D.1. Note because these criteria were not automatically checked, some runs may have run for slightly longer and therefore incurred a slightly higher cost. We observed that in practice this extra amount of time did not influence or change any of the results.

Table 8: API cost of each run on the VideoGameBench `test` split from Table 2 in USD($). Note that some runs were ended early when the agent was stuck or making no meaningful progress. Certain models are also significantly cheaper per token than others. The locally-hosted Qwen models do not have associated API costs.

| VideoGameBench | GPT-4o | Sonnet 3.7 | Gemini 2.5 Pro | LLaMA 4 | Gemini 2.0 Flash |
|---|---|---|---|---|---|
| Civilization I | $14.46 | $30.00 | $5.29 | $0.42 | $10.05 |
| The Need for Speed | $0.50 | $0.62 | $0.38 | $0.04 | $0.08 |
| The Incredible Machine | $1.36 | $5.26 | $1.31 | $0.11 | $0.18 |
| Pokemon Crystal | $15.04 | $29.64 | $4.78 | $0.14 | $3.88 |
| Doom II | $0.17 | $0.40 | $3.79 | $0.03 | $0.01 |
| Kirby's Dream Land | $8.91 | $2.98 | $3.89 | $0.09 | $0.16 |
| Link's Awakening (DX) | $7.86 | $20.00 | $18.51 | $0.09 | $0.18 |

### D.3 VARIANCE ANALYSIS FOR MAIN EXPERIMENTS

Using the best performing agent on VideoGameBench (Gemini 2.5 Pro), we re-run experiments five times for *Kirby's Dream Land*, which is the only game where the agent reaches a checkpoint (it completes $4.8\%$ of the game). We also re-run experiments five times for *Doom II* and *The Incredible Machine*, where the agent had no progress. In five runs, the sample variance on *Doom II* and *The Incredible Machine* is 0, while for *Kirby's Dream Land* it is $0.2$.

### D.4 ESTIMATING "EXACT" PROGRESS ON MAIN EXPERIMENTS

Many models achieve a score of "0%" on most games in VideoGameBench and VideoGameBench-Lite due to our coarse-grained scoring scheme of rewarding major checkpoints. To justify these scores and our qualitative analysis, we provide a finer-grained scoring scheme by manually matching the furthest frame or location that a model was able to achieve to when a player in the walkthroughs used in Table 7 reached that point in the game. Below, we show the "exact" estimated progress of the experiments in Section 4.

Table 9: We take one of the best performing models on VideoGameBench, Gemini 2.5 Pro, and re-run it five times on *Kirby*, which is the only game where it reached at least one checkpoint. We also run it five times on two games, *Doom II* and *The Incredible Machine*, where it had no progress.

| Gemini 2.5 Pro on VideoGameBench | Run 1 | Run 2 | Run 3 | Run 4 | Run 5 |
|---|---|---|---|---|---|
| Kirby's Dream Land | 1 | 0 | 1 | 1 | 1 |
| The Incredible Machine | 0 | 0 | 0 | 0 | 0 |
| Doom II | 0 | 0 | 0 | 0 | 0 |

Table 10: We report the exact estimated performance on VideoGameBench `test` split (not including secret games). Each score is displayed as as a percentage of the furthest game state that the agent achieved, manually estimated based on exactly where this state appears in the walkthrough videos described in Table 7. Civilization I in particular is very noisy due to the randomness of each game.

| VideoGameBench | GPT-4o | Sonnet 3.7 | Gemini 2.5 Pro | LLaMA 4 | Gemini 2.0 Flash | QwenVL-2.5 7B | QwenVL-2.5 32B |
|---|---|---|---|---|---|---|---|
| Civilization I | **1.99%** | 0.91% | 0.18% | 1.02% | 0% | 0.18% | 0.91% |
| The Need for Speed | **0.67%** | 0.07% | 0.39% | 0.07% | 0.52% | 0.00% | 0.02% |
| The Incredible Machine | 0.00% | **0.10%** | 0.00% | 0.00% | 0.00% | 0.00% | 0.00% |
| Pokemon Crystal | **1.04%** | 0.37% | 0.37% | 0.37% | 0.48% | 0.37% | 0.26% |
| Doom II | 0.12% | 0.13% | **0.17%** | 0.12% | 0.12% | 0.12% | 0.12% |
| Kirby's Dream Land (DX) | 2.89% | **5.07%** | **5.07%** | 2.45% | 2.89% | 2.10% | 2.89% |
| Link's Awakening (DX) | 0.65% | **0.85%** | **0.85%** | 0.65% | 0.65% | 0.65% | 0.65% |
| Overall Score | 1.05% | **1.07%** | 1.00% | 0.67% | 0.67% | 0.49% | 0.69% |

Table 11: We report the exact estimated performance on VideoGameBench Lite `test` split (not including secret games), where the game emulator pauses while waiting for a model's action. Each score is displayed as as a percentage of the furthest game state that the agent achieved, manually estimated based on exactly where this state appears in the walkthrough videos described in Table 7. 100% indicates a completed game.

| VideoGameBench Lite | GPT-4o | Sonnet 3.7 | Gemini 2.5 Pro | LLaMA 4 | Gemini 2.0 Flash | QwenVL-2.5 7B | QwenVL-2.5 32B |
|---|---|---|---|---|---|---|---|
| Doom II | 0.24% | **0.34%** | 0.18% | 0.12% | 0.13% | 0.12% | 0.17% |
| Kirby's Dream Land | 5.07% | 5.07% | **5.42%** | 2.45% | 2.89% | 2.89% | 3.50% |
| Link's Awakening (DX) | 0.67% | 0.81% | **0.85%** | 0.65% | 0.65% | 0.65% | 0.65% |
| Overall Score | 1.99% | 2.07% | **2.15%** | 1.07% | 1.22% | 1.22% | 1.44% |

# E    QUALITATIVE ANALYSIS

We provide specific examples of trajectories taken when generating Table 2 and Table 3. These examples provide qualitative evidence to support claims made in § 4.2. Many of the examples below can be attributed to a combination of issues, but we highlight the specific failure mode described with example(s).

## E.1    THE KNOWING-DOING GAP.

Similar to the findings in Paglieri et al. (2025), we found that models often struggle to execute the correct actions despite broadly knowing what to do.

In the example below, we show how a frontier VLM (Llama 4 Maverick) writes out what it should be doing (e.g. moving down towards the door to exit the room and explore outside) but it repeatedly presses "down" on the agent without aligning it actually face the door. In the rest of the trajectory, the agent continually repeats this action and is stuck on the same screen.

Furthermore, there are several reasons that suggest that the issue is not just the model's inability to perceive the screen. Firstly, the VG-Agent scaffold has access to all previous frames and actions, and can see that the screen is not changing despite it attempting to move out the room – even a human with poor perception would recognize that the door must be somewhere else if the screen has not changed in multiple steps. Secondly, in this particular example, the agent actually acknowledges earlier that the door is to the bottom-left of the character, but still chooses to "move down" to reach the door.

**Llama 4 Maverick playing *The Legend of Zelda: Link's Awakening* (VideoGameBench). Step 130**

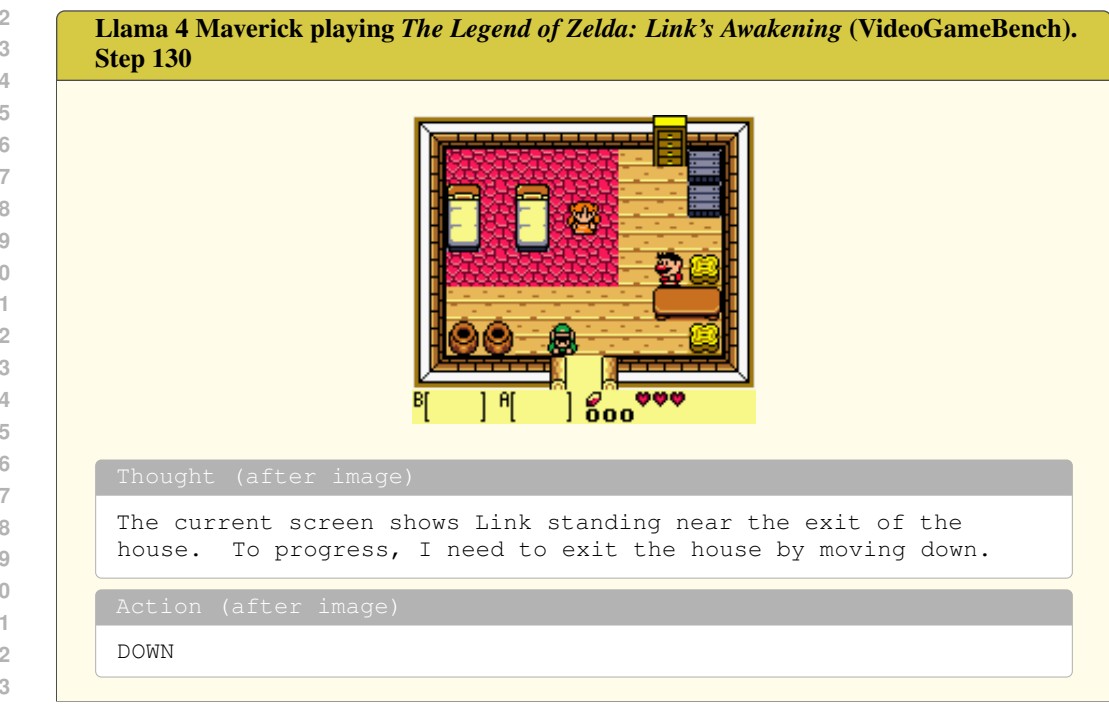

Thought (after image)

The current screen shows Link standing near the exit of the house.  To progress, I need to exit the house by moving down.

Action (after image)

DOWN

### E.2  FRONTIER MODELS STRUGGLE TO CORRECTLY PROCESS VISUAL INPUTS.

The following example shows an example of the model struggling to understand the screen. After clearing the starting room on the first level of *Doom II*, the Gemini 2.0 Flash VG-Agent continues firing at the enemies, and outputs that it thinks "the enemy is still alive. I need to adjust aim again and fire." In this trajectory, the agent eventually wastes all of its ammo and gets stuck in the first room.

**Gemini 2.0 Flash on Doom II (VideoGameBench Lite). Step 305**

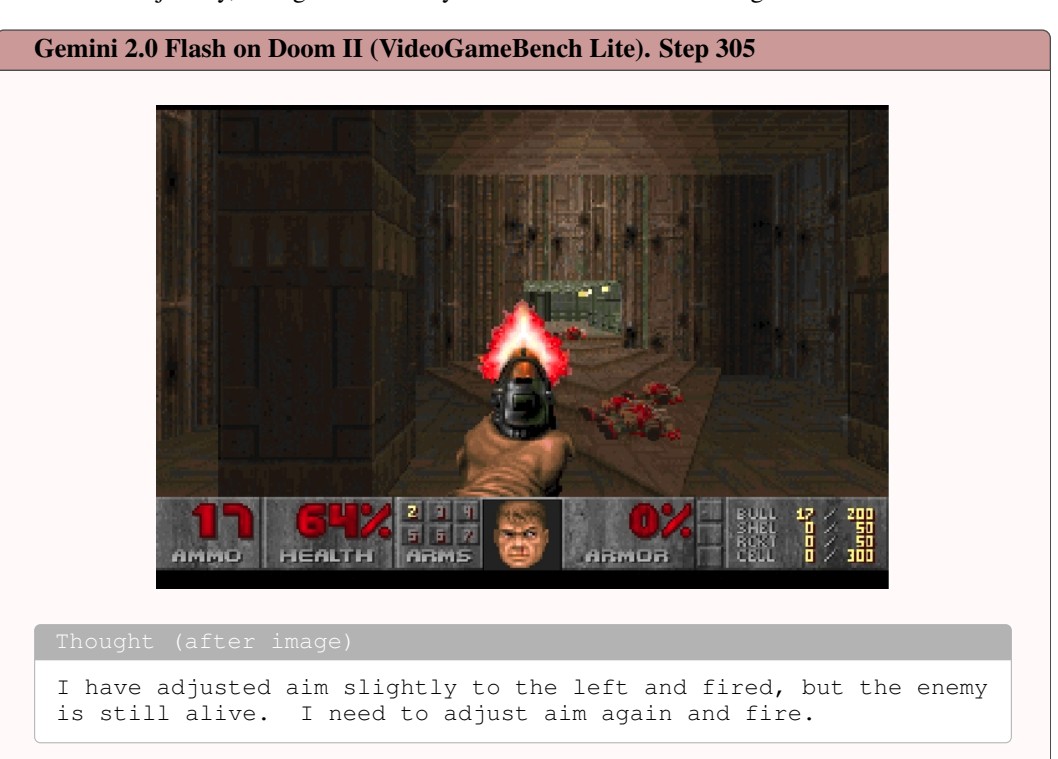

Thought (after image)

I have adjusted aim slightly to the left and fired, but the enemy is still alive.  I need to adjust aim again and fire.

In the next example, GPT-4o gets confused trying to interact with an NPC in *The Legend of Zelda*.
In this room, the agent must first talk to the man on the right before they can leave the room. After
exploring the starter room, the agent realizes it needs to talk to the NPC to grab a shield item. It sees
that it is near the NPC and continues to believe that it is interacting with the NPC despite no dialogue
appearing. It continues this behavior until later attempting to exit after falsely believing it has spoken
to the NPC.

**GPT-4o on *The Legend of Zelda: Link's Awakening* (VideoGameBench). Step 309**

Thought (after image)

**Current Situation and Strategy:**

- Link is still beside the NPC. No visual confirmation of the
shield yet.
- Primary goal remains to acquire the shield from the NPC.
- Continuing to engage with the NPC is necessary.

**Next Steps:**

1. **Press "A" repeatedly** to complete the dialogue.
2. **Ensure the shield is obtained and visible** in Link's
inventory.
3. Once the shield is confirmed, plan to exit the house.

Reflection (after image)

- Must confirm shield acquisition before leaving the house.  -
Pressing "A" multiple times ensures the dialogue is complete.
- Next objective is to explore outside once the shield is
obtained.  - Focus on completing the dialogue and shield
acquisition.  - Ensure shield acquisition is confirmed visually
before proceeding.  - Once confirmed, plan the path to exit the
house.  - Current priority is obtaining the shield from the NPC.

```
Action (after image)

"A",  # Initiate or continue dialogue.
"A",  # Ensure dialogue completion.
"A"   # Final confirmation for shield.
```

### E.3 VG-AGENT LACKS PLANNING AND MEMORY MANAGEMENT ABILITIES

We first expand on the example provided in the main paper, and also provide a visually obvious example below. In the *Doom II* on VideoGameBench Lite example, we found that all agents did not explicitly track or map out their surroundings. For agents that survived all the enemies, they often repeatedly walked back and forth between the starting area and first corridor, never making it to the ending area. For context, the first checkpoint / level of *Doom II* only requires taking two turns down a hallway and was completed by a co-author with no prior game experience in under 20 seconds.

We show another qualitative example below of *GPT-4o* on *Pokemon Crystal* forgetting that it had collected its first Pokemon after the first checkpoint, as well as forgetting its objective was to run an errand. It overwrites its textual scratch pad memory (the reflection) for a local objective, forgetting what it was trying to do.

---

**GPT-4o on *Pokemon Crystal* (VideoGameBench). Step 698**

```
Thought (before image)

1. **Current Situation**: Still trying to exit the lab.
2. **Objective**: Leave the lab to head towards Mr. Pokémon's
house.
3. **Analysis**: Ensure proper alignment with the exit carpet and
attempt to move down.
```

```
Reflection (before image)

- Received Cyndaquil as my starter Pokémon.
- Skipped the nickname option to proceed quickly.
- Objective: Visit Mr. Pokémon as directed by PROF. ELM.
- Need to exit the lab and start the journey towards Mr.
Pokémon's location.
- Route to Mr. Pokémon goes through Cherrygrove City.
- Learned to heal Pokémon with the lab machine if needed.
- Currently finishing dialogue with the lab assistant.
- Received a Potion from the assistant, which is now in the
ITEM POCKET.
- Currently engaged in a dialogue about the lab staff being busy.
- Attempting to exit the lab to begin the journey.
```

```
Thought (after image)

**Step-by-Step Plan:**

1. **Current Situation**: Stuck at the lab exit.
2. **Objective**: Exit the lab and start the journey to Mr.
Pokémon's house.
3. **Analysis**: Confirm alignment with the exit and check
potential obstacles.
```

```
Reflection (after image)

- Need to ensure proper alignment and navigation to exit
buildings.
- Current focus is on successfully leaving the lab to progress
the storyline.
- Stuck at the exit requires paying attention to visual cues for
proper movement.
```

```
Action (after image)

UP, DOWN, DOWN
```

## F  PRACTICE GAMES

In this section, we expand on our three practice games defined in § 4 and Table 4. We also verify that each game is solvable by humans playing through the VideoGameBench interface.

### F.1  LOCATION CLICKING GAME

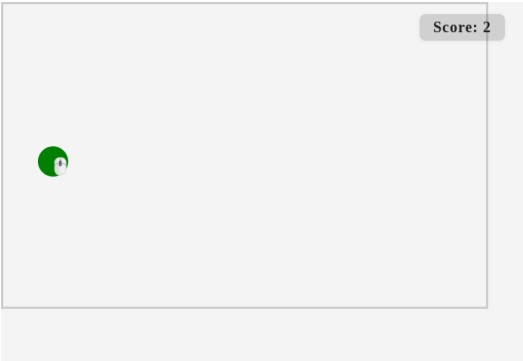

Figure 6: An example screen of the *Location Clicking Game*. A VG-Agent using a VLM is tasked with clicking 10 green circles, one at a time, in under 250 actions.

The most basic action in any DOS game is to click a position on the screen. The *Location Clicking Game* is a simple task where an agent must click a green circle with radius 40px that randomly generates inside a 640px by 400px region on the browser (this replicates the DOS game setting on VideoGameBench that runs inside a 640px by 400px window). Each time the agent clicks the circle, it respawns in a new location, and the agent must click the circle 10 times in under 250 actions. While some VLMs such as Claude Sonnet 3.7 and Gemini 2.5 Pro do not struggle with this task, many frontier VLMs require multiple correction steps to click the circle, and cannot complete the game in under 250 actions.

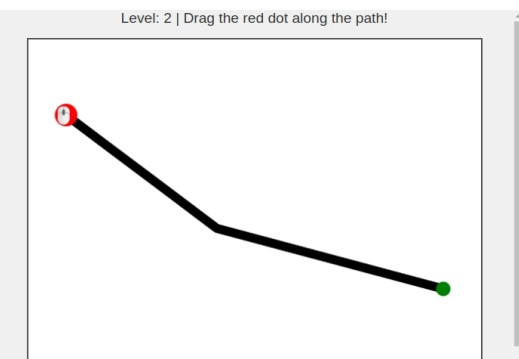

Figure 7: An example screen of the *Mouse Dragging Game*. A VG-Agent using a VLM is tasked with dragging a red circle to a target green circle while staying on the black line.

### F.2 DRAGGING GAME

The VideoGameBench mouse and keyboard interface currently supports dragging the mouse from a position $(x_0, y_0)$ on the screen to another position $(x_1, y_1)$ in a straight line. Many games in VideoGameBench such as *The Incredible Machine* and *Age of Empires* require or allow dragging the mouse to play the game. On the *Mouse Dragging Game*, we devised 10 simple challenges for an agent to drag in a certain pattern. We find that all VLMs struggle on this challenge, with only Claude Sonnet 3.7 even completing the first level, which is just to drag in a straight horizontal line.

### F.3 2D NAVIGATION GAME

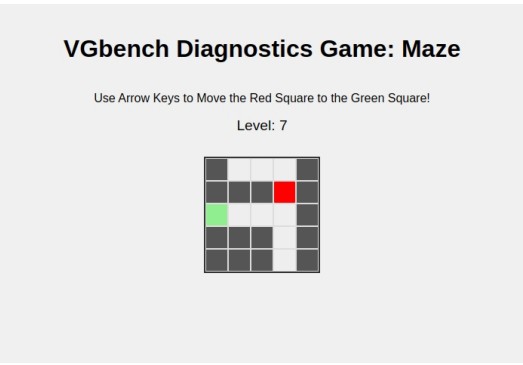

Figure 8: An example level in the *2D Navigation Game*. The task is to use the arrow keys (mouse + keyboard interface on VideoGameBench) to move a red square to the green square in a small grid-world maze.

In many DOS and Game Boy games in VideoGameBench, the agent must move a character in a 2D (often grid-world, e.g. *Pokemon Crystal*) environment. While this task is often easy for specialized agents, for VLMs it is not obvious that this task is easily solved. We generate 10 pre-defined mazes where the agent must move a red square to the green square in a small maze-like environment using the arrow keys. There are movable tiles (light gray) and immovable tiles (dark gray) that the agent must navigate. Although each maze can be solved in under 10 steps, we find that no agent can complete the entire game in under 250 actions, i.e. 25 steps per maze.

