# OpenReview forum: "VideoGameBench: Can Vision-Language Models complete popular video games?"
_ICLR.cc/2026/Conference — ICLR 2026 Conference Desk Rejected Submission_

### Official Review · Reviewer_2JDe · 2025-10-26

**Soundness:** 2
**Presentation:** 3
**Contribution:** 2
**Rating:** 2
**Confidence:** 5

**Summary:**

The paper presents a benchmark for Vision Language Models on a variety of popular video games from the 90s. Most current models get 0% performance on basically every single task with this setup.

**Strengths:**

The benchmark is very cool, and the engineering and effort that must have gone behind making this are surely impressive. The paper is well written and easy to follow.

**Weaknesses:**

One of my main criticisms for this paper is that a benchmark were all the current models score 0% on basically every single task is not an interesting benchmark. This surely makes the benchmark future-proof, but the amount of insights it can provide now compared to existing benchmarks with more fine-grained progression systems is lacking. Without a more fine-grained progression system, or easier games where current VLMs can get some amount of performance, the benchmark can offer limited insights compared to what has already been shown in related work, regardless of how cool it is.

I am not sure this benchmark is asking the right questions. Do we expect VLMs to be the right format/models for these games? I fully get the reason to test VLM on games requiring reasoning, but a good chunk of these games mostly require accurate perception-to-muscle memory and fast reflexes for humans to perform well, skills that we are likely decades away from unlocking in general purpose multimodal language models. Perhaps Vision-Language-Action (VLA) models, more akin to what is being developed for robotics are the better test target for this benchmark. Or perhaps something more similar to SIMA [SIMA Team, 2024], (which I think would deserve a citation in this paper) would be more appropriate for video games. Or perhaps, the subset of video games chosen for the benchmark should be reconsidered based on what models are being tested.

The latency part of this benchmark (full version, not lite) also feels strange, as latency on most of these models is inherently dependent on the GPU being used and/or current server traffic for proprietary models, so a big variable is very much out of your control. At that point you are really benchmarking the quality of the provided service at that point in time and not necessarily the quality of the models themselves. I’m sure with a proper set up and the right amount of compute you could run most present-day models in real time with no latency problems, but this is obviously difficult to achieve if you are not the provider yourself. What happens if in the future you change the GPUs you're running with, or if the API you are using is undergoing more/less traffic?

Benchmarks for LLM/VLMs on games are inherently high-variance. One seed is unfortunately not enough.

In the current state, this benchmark, while extremely cool and impressive in its own right, seems to be missing a target (the right type of foundation model to be tested) and a purpose (what new questions is it trying to answer? What new insights is it providing?)

**Questions:**

1. Why weren’t easier games added to get a more fine-grained average progression?
2. What new insights is this benchmark providing in its current state?
3. Why are VLMs the right target for this benchmark and not VLAs?
4. Using only a frame every 0.5 seconds doesn't seem enough for many of these games. Have other frame rates been tested?
4. Why is latency an important factor here, given that it’s something out of your control for most of the close-source models? What GPUs did you use to test open-source models? How do you account for changes in latency of close-source models APIs?

---

> ### Author Response · Authors · 2025-12-02
> **Response to strengths, weaknesses, and comments**
>
> We thank the reviewer for taking the time to read and write valuable feedback for our paper. We also appreciate the reviewer’s acknowledgement of the benchmark and the paper’s writing.
>
> We answer your specific concerns below:
>
> > One of my main criticisms for this paper is that a benchmark where all the current models score 0% on basically every single task is not an interesting benchmark.
>
> We explicitly address this in Section 4.1’s “Interpreting “0%” scores on games”, where we point to Tables 10 & 11, which mark the “true” progress of each game (being non-zero for almost all games).
>
> We also want to add that there are many examples of widely used benchmarks (e.g. SWE-Bench [1] which started at 1.96% and is now >70%, Humanity’s Last Exam [2] which started at 3.3% and is now at 25.4%) that also reported near zero scores from their inception, but saw quick improvements over the past year. We argue that current models scoring low but showing promise on practice games and on several different games in both the real-time and Lite settings is not a weakness, but rather a strength of the benchmark.
>
> > Do we expect VLMs to be the right format/models for these games? I fully get the reason to test VLM on games requiring reasoning, but a good chunk of these games mostly require accurate perception-to-muscle memory and fast reflexes for humans to perform well, skills that we are likely decades away from unlocking in general purpose multimodal language models. Perhaps Vision-Language-Action (VLA) models… are the better test target for this benchmark.
>
> VideoGameBench is intended to **measure the progress of VLMs** on realistic video games with minimal human intervention – these allow for testing VLMs on interactive tasks that crucially require both visual and language processing skills. The goal is not to test whether automated systems can play games – we have shown that this is possible with RL methods like DQN and PPO in much harder settings like Starcraft and Dota 2. VLAs, which we also discuss in the related works, are a separate, game-specific method that likely also can play many of these games. We are interested in understanding whether general purpose visual-language learners (e.g. VLMs) are able to generalize well to settings that are intuitive for humans to solve. Lastly, we purposely added VideoGameBench-Lite to isolate the latency constraint of these models, so “accurate perception-to-muscle-memory” is not needed to solve these tasks.
>
> > Or perhaps something more similar to SIMA [SIMA Team, 2024], (which I think would deserve a citation in this paper) would be more appropriate for video games.
>
> We point out that SIMA was already cited and discussed in the Related Works section 5.1.
>
> > Benchmarks for LLM/VLMs on games are inherently high-variance. One seed is unfortunately not enough.
>
> We agree that the performance of LMs / VLMs on benchmarks are inherently high variance. We provide a variance experiment in Appendix D.3 for Gemini 2.5 Pro on Kirby (which achieves a non-zero score), but we generally find that these models are often unable to meaningfully progress in many games, leading to them failing at the same point. Unfortunately, while we would like to add more experiments, it is out of the scope of our budget to repeat these experiments several times across all games.
>
> [1] Jimenez et al. “SWE-bench: Can Language Models Resolve Real-world Github Issues?” The Twelfth International Conference on Learning Representations, 2024.
>
> [2] Phan et al. “Humanity’s Last Exam.” arXiv preprint arXiv:2501.14249, 2025.

---

> ### Author Response · Authors · 2025-12-02
> **Response to specific questions**
>
> > 1. Why weren’t easier games added to get a more fine-grained average progression?
>
> We added the practice games in our analysis to isolate specific failure modes of each model. However, given the existence of other game benchmarks for VLMs with partial progress (e.g. BALROG [1]), we focused VideoGameBench (and Lite) on a suite of challenging games from different genres to add a new benchmark that challenges frontier VLMs.
>
> > 2. What new insights is this benchmark providing in its current state?
> It is not obvious that frontier models would perform poorly on VideoGameBench-Lite, where latency concerns from models (i.e. thinking time) are removed because the game is paused after every action is taken. These settings are analogous to Gym environments, where models, including VLMs, have seen success. This benchmark shows that latency is a large issue, but beyond latency, existing frontier VLMs are unable to reason successfully on games where humans intuitively are able to solve them. This finding is contrary to many existing findings that VLMs and LMs have been able to display super-human performance across many different tasks. Our paper provides the insight that there are several failure modes of existing VLMs on these settings that challenge them as general-purpose vision language solvers.
>
> > 3. Why are VLMs the right target for this benchmark and not VLAs?
>
> We agree that a video game benchmark targeted towards VLAs is valuable – even the VideoGameBench suite for VLAs is valuable. We are particularly interested in understanding and pushing the limits of general purpose solvers, and VLMs are able to solve tasks beyond games, unlike VLAs. As discussed in Rebuttal Point 2, the purpose of VideoGameBench is not to understand whether games can be solved by agents, but rather if a general purpose VLM can reason about a video game. The real-time component is independently important in understanding whether these systems can be deployed in real-time settings.
>
> > 4. Using only a frame every 0.5 seconds doesn't seem enough for many of these games. Have other frame rates been tested?
>
> We found in practice through our small human study described in **Appendix C.2** (i.e. have a human play the game under these constraints) that providing a frame every 0.5 seconds is sufficient for a human to make significant progress with minimal effort on many of the games. We tested other frame rates, but found that they quickly fill up the context window of modern frontier models, making them consistently worse on each game. We did not rigorously test performance across different frame rates, but we also found qualitatively in Appendix E that none of the failure modes were due to missing information from finer-grained frames.
>
> > 5. Why is latency an important factor here, given that it’s something out of your control for most of the close-source models? What GPUs did you use to test open-source models? How do you account for changes in latency of close-source models APIs?
>
> Latency is one of the most important distinguishing factors between answering whether a VLM can reason about and solve a game given unlimited time (purely a reasoning problem), and whether a VLM can just solve a game (a reasoning + latency problem). However, before latency is even a concern, the ability for a model to reason through and solve a game is necessary. We did not extensively account for variance in the latency of closed source APIs or effects like GPU throttling, but the poor performance on VideoGameBench-Lite suggests that the performance gap of frontier models is due to an ability to reason / process game frames correctly, rather than variance in model inference time.
>
> [1] Davide Paglieri et al. “BALROG: Benchmarking agentic llm and vlm reasoning on games.” The Thirteenth International Conference on Learning Representations, 2025.

---

### Official Review · Reviewer_93zX · 2025-11-01

**Soundness:** 3
**Presentation:** 3
**Contribution:** 3
**Rating:** 4
**Confidence:** 4

**Summary:**

The paper introduces VideoGameBench, a benchmark for evaluating vision-language models (VLMs) on 23 curated 1990s video games (split into dev/test sets, including 3 secret games) from Game Boy and MS-DOS platforms. Unlike existing benchmarks, it requires VLMs to complete full games in real time using only raw visual inputs and high-level objectives/controls (no game-specific scaffolding or auxiliary tools). The authors also propose VideoGameBench Lite, a variant that pauses the game during model inference to mitigate latency issues.

**Strengths:**

1. The paper addresses a critical underexplored gap: evaluating VLMs on full, unmodified real-world tasks (1990s video games) that require integrated human-like abilities (perception, memory, real-time decision-making). Prior benchmarks rely on simplified grid worlds, text-only games, or game-specific tools (e.g., Gemini Plays Pokemon used pathfinding hints), making VideoGameBench a novel "no crutches" evaluation.
2. The benchmark construction is rigorous: it supports multiple emulators (PyBoy, DOSBox), standardizes action interfaces (keyboard/mouse/controller), and uses human-verified checkpoints scraped from walkthroughs. The automated progress-tracking via perceptual hashing is well-validated (e.g., tuning Hamming distance thresholds per game) and avoids subjective manual scoring.
3. The work highlights critical limitations of state-of-the-art VLMs in real-world, dynamic environments—limitations that matter for downstream applications (e.g., robotics, autonomous systems). For example, the "knowing-doing gap" and poor memory management are not captured by math/coding benchmarks but are essential for embodied AI.
4. VideoGameBench provides a standardized, scalable testbed to drive progress in VLM generalization and real-time reasoning. By focusing on 1990s games (with well-documented mechanics and walkthroughs), it avoids copyright or accessibility barriers of modern games while retaining complexity.

**Weaknesses:**

1. The benchmark focuses exclusively on 1990s 2D/3D games (Game Boy, MS-DOS), excluding modern game mechanics (e.g., open-world exploration, multiplayer, touch controls) or other classic platforms (e.g., NES, Sega Genesis). This narrows the generalizability of results—VLMs may fail differently on games with distinct interaction paradigms (e.g., point-and-click adventures vs. real-time strategy).
2. There is no human baseline for the full benchmark—only confirmation that humans can complete practice games and first checkpoints. A human performance metric (e.g., average completion percentage, time to checkpoint) would better contextualize VLM failure.

**Questions:**

1. The paper mentions that "0% score does not imply no progress"—but how much incremental progress (e.g., moving 1 tile, interacting with 1 object) do models typically make? Providing finer-grained scores for all games (not just Table 10–11) would better illustrate model capabilities.
2. Did you test if VLMs perform better with game-specific prompts beyond basic controls/objectives (e.g., hints about common mechanics like "jump to avoid enemies")? If not, could prompt engineering reduce the "knowing-doing gap"?
3. More model results are needed to include. This paper only provides 7 VLMs.

---

> ### Author Response · Authors · 2025-12-02
> **Response to questions on additional experiments and fine-grained scoring**
>
> Thank you for taking the time to review this work. We appreciate that the reviewer recognizes the **value of evaluating VLMs on real-world video game tasks** that require human-like skills, as well as the **robustness of our automated evaluation framework**.
>
> We address your specific concerns below:
>
> > The benchmark focuses exclusively on 1990s 2D/3D games (Game Boy, MS-DOS), excluding modern game mechanics… This narrows the generalizability of results—VLMs may fail differently on games with distinct interaction paradigms.
>
> We agree that adding more games across other time periods / consoles would allow for testing more generalization capabilities. However, in the case of VideoGameBench, we already provide a suite of 10 test games across different genres to evaluate generalization to different domains with distinct interaction paradigms (e.g. racing vs. RTS vs. FPS). This follows from other related recent papers that have appeared in ICLR, such as BALROG [1] which benchmarked 6 different games.
> > There is no human baseline for the full benchmark—only confirmation that humans can complete practice games and first checkpoints. A human performance metric (e.g., average completion percentage, time to checkpoint) would better contextualize VLM failure.
>
> A full human study on average completion time per checkpoint and completion rate would add to the study, but is beyond the scope of this paper. The existence of online human walkthroughs of each game already proves that humans can complete each game, and they provide noisy estimates of the completion time and capabilities of human players.
>
> > The paper mentions that "0% score does not imply no progress"—but how much incremental progress (e.g., moving 1 tile, interacting with 1 object) do models typically make? Providing finer-grained scores for all games (not just Table 10–11) would better illustrate model capabilities.
>
> **Tables 10-11** provide the finest grain progress for each game – they score based on the furthest possible point in a walkthrough (e.g. moving 1 extra tile in Kirby) that the model achieves. We do not explicitly write where in each game each model reaches (this can be inferred from Tables 10-11), but we provide all trajectories in our supplementary which contain the exact points where the models failed. Finally, the amount of incremental progress is noisy and depends on the particular checkpoint and game, and we provide explicit checkpoints and trajectories in Section 4.2 and Appendix E that highlight the exact points where certain models fail.
>
> > Did you test if VLMs perform better with game-specific prompts beyond basic controls/objectives (e.g., hints about common mechanics like "jump to avoid enemies")? If not, could prompt engineering reduce the "knowing-doing gap"?
>
> We did not extensively study the effect of prompt engineering on VideoGameBench, but we agree that this would add to the analysis. We primarily focused this work on understanding how well general-purpose VLMs would perform on video games without extra prompting beyond basic game instructions, and we leave this as future work.
>
> > More model results are needed to include. This paper only provides 7 VLMs.
>
> We follow existing works like BALROG [1] and SWE-Bench Multimodal [2] that evaluate across ~7 frontier VLMs in their main experiments. Our experiments evaluate VideoGameBench and VideoGameBench-Lite on most frontier VLMs (GPT-4o, Claude, Gemini 2.5 Flash / Pro) as well as popular open source VLMs (Llama 4, QwenVL), which we found to be sufficient in uncovering common failure modes across both top closed-source and open-source models.
>
> [1] Davide Paglieri et al. “BALROG: Benchmarking agentic llm and vlm reasoning on games.” The Thirteenth International Conference on Learning Representations, 2025.
>
> [2] John Yang et al. “SWE-Bench Multimodal: Do AI Systems Generalize to Visual Software Domains?” The Thirteenth International Conference on Learning Representations, 2025.

---

### Official Review · Reviewer_PQJm · 2025-11-01

**Soundness:** 2
**Presentation:** 3
**Contribution:** 2
**Rating:** 2
**Confidence:** 5

**Summary:**

Evaluate whether vision-language models (VLMs) can perform human-like embodied reasoning by playing full video games from the 1990s using only visual input and minimal instructions. Experiments show all models fail to exceed the start of any game.

**Strengths:**

- Challenging environments and strict train/test split are well-needed.
- Use of perceptual hash to estimate game completion is interesting.

**Weaknesses:**

- Many similar benchmarks already exist, as the author's have cited.
- Evaluation scope narrow: experiments limited to a handful of VLMs; no systematic scaling or modality ablation.

**Questions:**

- Can the author's correlate and report perceptual hash with some ground truth measure of the progress in a game, i.e. using the walkthrough videos?
- Additionally, how does perceptual hash perform for games where progression is not entirely linear? E.g. The Legend of Zelda.
- Given the diverse skills required for each game, could the games be categorised such that the benchmark measures multiple discrete capabilities of models?

**Details Of Ethics Concerns:**

While the use of open-source emulators (e.g., PyBoy) is legal, most of the referenced games remain under copyright protection. The author's briefly address this, however do not clarify on what the benchmark itself distributes. In practice, reproducing the experiments would require accessing copyrighted ROMs, which cannot be legally distributed. Therefore, I am a bit concerned about the legality of this work.

---

> ### Author Response · Authors · 2025-12-02
> **Response to questions on benchmarking and scoring**
>
> Thank you for taking the time to review. We appreciate that the reviewer acknowledges the **value of the checkpointing system**.
>
> We address your specific concerns below.
> > Many similar benchmarks already exist, as the author's have cited.
>
> In our paper, we cited several benchmarks that consider game-like settings for VLMs, which the reviewer is referring to. However, we also explicitly highlight throughout the paper (see Section 1, Section 2.1, Section 5) and in the Related Works section how each of these benchmarks differ in both their tasks and goal, which we re-iterate below. BALROG [1] is an existing VLM benchmark that focuses on grid-world or text-only settings for games. LMGameBench [2] is a VLM benchmark that contains real-world games, but does not emphasize or distinguish between real-time and non-real-time games. GameTraversalBench [3] is an LM benchmark that tracks an LMs ability to plan in maze-like environments.
>
> To our knowledge, there are no benchmarks that focus on a modern suite of long horizon video games from for VLMs across a wide range of genres. Furthermore, the VideoGameBench / VideoGameBench-Lite split is the first to separately evaluate models based on and without latency constraints.
>
> > Evaluation scope narrow: experiments limited to a handful of VLMs; no systematic scaling or modality ablation. We evaluate VideoGameBench and VideoGameBench-Lite on most frontier VLMs (GPT-4o, Claude, Gemini 2.5 Flash / Pro) as well as popular open source VLMs (Llama 4, QwenVL), which we found to be sufficient in uncovering common failure modes across all models. We follow existing works like BALROG [1] that evaluate across ~7 frontier VLMs.
>
> We agree that evaluating models across different scales is interesting, but given the poor performance of the top frontier models on both VideoGameBench and VideoGameBench-Lite, we opted out of evaluating smaller models. In addition, for frontier models, in most cases smaller versions of those models are not publicly available. To provide a spectrum for general video game playing capabilities of each model, we also isolate specific failure modes of models using the practice games (see Section 4), which provide key initial areas of improvement for VLMs. For modality ablation, the focus of VideoGameBench is to test how VLMs perform when given visual frames on the raw game, rather than using extra information.
>
> > Can the author's correlate and report perceptual hash with some ground truth measure of the progress in a game, i.e. using the walkthrough videos?
>
> In Section 2.5, we describe the scoring mechanism, which is purely based on how far an agent reaches in the game relative to a fixed online walkthrough. Perceptual hashing itself is a method for matching in-game frames to existing checkpoint frames to make this process automatic during agent evaluation, but does not affect the ground truth measure of progress in game. The walkthrough videos themselves are directly used as a ground truth signal for measuring progress.
>
> > Given the diverse skills required for each game, could the games be categorised such that the benchmark measures multiple discrete capabilities of models?
>
> The games in VideoGameBench are categorized according to genre (e.g. platformer, real-time strategy, racing, etc.) in Table 1, which implicitly provides a categorization of the skills needed to beat the game. For each game in the test set, we also provide a corresponding game in the same genre in the train set. Nevertheless, beyond a loose categorization, provably identifying the exact skills per game is beyond the scope of this paper.
>
> > The author's briefly address this, however do not clarify on what the benchmark itself distributes. In practice, reproducing the experiments would require accessing copyrighted ROMs, which cannot be legally distributed. Therefore, I am a bit concerned about the legality of this work.
>
> We agree with the reviewer that copyright concerns must be carefully addressed in this work, which we have discussed in Section 6 and the Ethics Statement. VideoGameBench provides a framework for loading and running VLMs on games and evaluating their progress, which does not fall under copyright protections. Furthermore, legally re-producing or expanding on experiments requires access to copyrighted ROMs, which can be legally obtained through purchasing. Lastly, as specified in Section 6, **we do not distribute the game ROMs** with the benchmark framework.
>
> [1] Davide Paglieri et al. “BALROG: Benchmarking agentic llm and vlm reasoning on games.” The Thirteenth International Conference on Learning Representations, 2024.
>
> [2] Lanxiang Hu et al. “lmgame-Bench: How Good are LLMs at Playing Games?” arXiv:2505.15146, 2025.
>
> [3] Muhammad Umair Nasir et al. “GameTraversalBenchmark: Evaluating planning abilities of large language models through traversing 2d game maps.” arXiv:2410.07765, 2024.

---

### Official Review · Reviewer_Lkc1 · 2025-11-12

**Soundness:** 2
**Presentation:** 2
**Contribution:** 2
**Rating:** 4
**Confidence:** 3

**Summary:**

# Summary
This paper introduces **VideoGameBench**, a multimodal, interactive benchmark encompassing 23 classic video games (10 for testing, 13 for development) across Game Boy and MS-DOS platforms. It adopts a stringent ruleset requiring "raw pixels + only control/objective descriptions". Complementary components include **VideoGameBench Lite** (where the game pauses during agent inference) and an automated **checkpoint-hashing** method for progress measurement—this method leverages frame data from YouTube walkthroughs to quantify completion. Frontier vision-language models (VLMs) achieve extremely low completion rates: the top-performing models reach approximately 0.48% on the full benchmark and 1.6% on VideoGameBench Lite. Additionally, simple auxiliary "practice games" reveal critical deficits in VLMs’ mouse control, dragging precision, and basic navigation abilities.


# Strengths
1. **Clear problem framing and motivation**: The paper persuasively argues that current VLMs poorly capture human-centric inductive biases (e.g., perception, spatial navigation, memory management)—capabilities that come naturally to humans. It further justifies real-world video games as a intuitive, ecologically valid testbed for evaluating these understudied skills.
2. **Rigorous ruleset**: The constraint of "no visual overlays, no access to internal game state, and minimal hints" cleanly isolates VLMs’ core visual understanding and action-sequencing abilities. This design discourages over-reliance on bespoke toolchains or game-specific workarounds, ensuring evaluations reflect generalizable capabilities.
3. **Turn-based ablation (VideoGameBench Lite)**: This variant effectively decouples reasoning performance from inference latency. It clearly demonstrates that even without real-time response pressure, VLMs still struggle to progress—highlighting fundamental limitations in reasoning rather than just timing constraints.
4. **Automated progress tracking**: The perceptual-hashing pipeline provides a practical, replay-agnostic approach to coarse-grained scoring. By grounding checkpoints in publicly available walkthroughs, it ensures consistency and reproducibility across different model evaluations.
5. **Broad coverage across genres and I/O modalities**: The benchmark spans diverse game mechanics (controller-based for Game Boy vs. mouse/keyboard for MS-DOS), spatial dimensions (2D and 3D), and genres (action, RPG, strategy, puzzle, racing). This breadth enables comprehensive assessment of VLMs’ adaptability to varied interactive tasks.


# Weaknesses
1. **Discussion of checkpoints as performance metrics**: While using checkpoints as a performance metric is intuitively reasonable, results in Table 2 show that most models score 0% on the majority of games. This excessively high benchmark difficulty makes it difficult to reflect models’ true capabilities and leads to a lack of discriminative power between different VLMs. Supplementing the checkpoint system with more granular, phased metrics to capture incremental model progress would significantly enhance the benchmark’s general applicability.
2. **Mechanistic failure analysis**: The identified failure modes (e.g., the "knowing-doing gap") are descriptive but lack mechanistic depth. For instance, the paper does not address: Is the knowing-doing gap caused by VLMs’ inability to map visual perception to action sequences, limitations in context window size (e.g., forgetting prior actions), or poor spatial coordinate mapping? Without such mechanistic insights, the paper provides limited guidance for targeted model improvement.


# Questions
Q1: Can the authors supplement the benchmark with phased, granular metrics to better capture incremental model capabilities? Doing so would address the current lack of discriminative power and further enhance the benchmark’s general applicability (targeting Weakness 1).
Q2: Can the authors conduct a more mechanistic analysis of the identified failure modes? For example, disentangling whether deficits stem from visual perception-action mapping, context window constraints, or spatial reasoning limitations would provide critical insights for model development (targeting Weakness 2).

**Strengths:**

See Summary

**Weaknesses:**

See Summary

**Questions:**

See Summary

---

> ### Author Response · Authors · 2025-12-02
> **Response to questions on scoring and extra analysis**
>
> Thank you for taking the time to review. We appreciate that the reviewer recognizes the **motivation and coverage** of VideoGameBench, as well as the **value of VideoGameBench Lite for isolating the latency constraint** of frontier models.
>
> We thank you for your feedback, and address your specific concerns below.
>
> > Discussion of checkpoints as performance metrics: While using checkpoints as a performance metric is intuitively reasonable, results in Table 2 show that most models score 0% on the majority of games. This excessively high benchmark difficulty makes it difficult to reflect models’ true capabilities and leads to a lack of discriminative power between different VLMs.
>
> In Appendix D.4, we provide a fine-grained scoring mechanism based on exact walkthrough progress on a particular game. **We show scores using this fine-grained scoring in Table 10**, which is derived from the results in Table 2 by tracking the furthest point of progress in a game (relative to the walkthrough) that each model makes rather than the furthest checkpoint.
>
> We want to highlight that the use of checkpoints rather than this fine-grained scoring mechanism offers a consistent and automatic mechanism for detecting progress in games rather than extra manual effort. We provided this extra analysis in the appendix to supplement the near zero scores and discuss this analysis in the sub-section, **“Interpreting “0%” scores on games.”**
>
> > The identified failure modes (e.g., the "knowing-doing gap") are descriptive but lack mechanistic depth. For instance, the paper does not address: Is the knowing-doing gap caused by VLMs’ inability to map visual perception to action sequences, limitations in context window size (e.g., forgetting prior actions), or poor spatial coordinate mapping?
>
> We explicitly provide the practice games in Section 4 and Table 4 to isolate specific failure modes of VLMs on simplified game settings. Poor performance on these games highlights initial areas of improvement for frontier VLMs. Furthermore, many of these specific failure modes are entangled and not exclusive to video games, which we discuss from prior works described in “Models frequently struggle to correctly process visual inputs.”.
>
> Paired with the performance of each model on each of the practice games, we can infer particular failure modes in the qualitative analysis section. For example, the lack of performance of GPT-5, Gemini 2.5 Pro, Llama 4, and QwenVL-2 on the clicking game suggests poor spatial coordinate mapping abilities, which translates to the Knowing-doing Gap. The purpose of VideoGameBench is to highlight these failure modes on a visual domain where humans succeed.
>
> > Q1: Can the authors supplement the benchmark with phased, granular metrics to better capture incremental model capabilities? Doing so would address the current lack of discriminative power and further enhance the benchmark’s general applicability (targeting Weakness 1).
>
> See Table 10 and the answer to Q1, which provides fine-grained metrics for all main experiments on VideoGameBench and VideoGameBench-Lite. We will revise the scoring of the main table (Table 2) to be Table 10, and instead opt for the more fine-grained scoring as the main quantitative result.
>
> > Q2: Can the authors conduct a more mechanistic analysis of the identified failure modes? For example, disentangling whether deficits stem from visual perception-action mapping, context window constraints, or spatial reasoning limitations would provide critical insights for model development (targeting Weakness 2).
>
> See answer to the second quoted response. We further note that the particular examples made here are directly addressed in the paper – visual perception-action mapping is the primary cause of the Llama 4 on Zelda example in Appendix E, context window constraints contribute to the difficulty of long-term memory, and spatial reasoning limitations contribute to the Kirby example in Appendix E. While we agree that more rigorous mechanistic analysis beyond what was provided is insightful, isolating each component is beyond the scope of this benchmark, which aims to evaluate VLMs on modern video games.

---

### Author Response · Authors · 2025-12-02
**Addressing general concerns and changes to the paper**

We appreciate the reviewers’ time and effort in providing valuable feedback. We have integrated the rebuttal content into  the paper to address the reviewers’ concerns, and have also directly responded to and addressed each reviewers’ individual concerns.

To re-iterate, **VideoGameBench**’s goal is to measure and direct the progress of VLMs, which have served as general-purpose models that can solve vision-language tasks, on a diverse set of interactive game settings with minimal scaffold engineering. Our main contributions are:
* **VideoGameBench benchmark:** A novel, challenging, realistic, and easy to understand benchmark that focuses on game-solving abilities. VideoGameBench offers a goalpost for evaluating VLM performance on interactive tasks.
* **De-coupling inference latency:** VideoGameBench-Lite is an alternative, gym-like environment to de-couple the inference latency bottleneck of most frontier models and isolate failure modes of VLMs on games. This separate benchmark also allows for evaluating model progress on games without the latency bottleneck.
* **Semi-automatic scoring system:** VideoGameBench scores progress on video games without an in-game metric. We introduce a novel scoring system based on matching game frames to checkpoints, while also assigning scores by referencing existing walkthrough timestamps.
* **Experiments & Qualitative Analysis:** We provide baseline performance for the state-of-the-art closed and open source VLMs. We also provide extensive qualitative analysis on specific trajectories where the models succeed or fail.

Some common points of concern we want to address are the following:

* **Understanding 0% scores on Table 2 vs. Table 10**. We found a lot of confusion regarding Table 2, which shows many models scoring 0% across many games, despite the paragraph in Section 4.1 explaining how these are coarse-grained, checkpoint-based scores. Table 10 is a fine-grained version of Table 2, which shows the finer differences between VLMs on each game. To avoid future confusion, we are moving Table 10 / 11 as the main results to show finer differences between each model.

* **On adding more VLMs to our experiments**. We have included independent experiments for 7 different frontier VLMs (4 closed, 3 open) across most model providers on all games in both VideoGameBench and VideoGameBench-Lite. We cite BALROG [1] and SWE-Bench Multimodal [2] as two existing VLM benchmarks previously accepted to ICLR that use seven or less of roughly the same subset of VLM families (older versions) to justify our breadth of experiments.

* **On adding more fine-grained analysis or games**. In addition to the fine-grained quantitative results for both VideoGameBench and VideoGameBench-Lite in Tables 10/11 respectively, we have included trajectories for all games to show where frontier models fail. Furthermore, we provide the practice games as evidence that all evaluated frontier VLMs struggle on basic game-playing skills, implying that a more complicated hierarchy of games would similarly show poor performance of existing models.

[1] Davide Paglieri et al. “BALROG: Benchmarking agentic llm and vlm reasoning on games.” The Thirteenth International Conference on Learning Representations, 2025.

[2] John Yang et al. “SWE-Bench Multimodal: Do AI Systems Generalize to Visual Software Domains?” The Thirteenth International Conference on Learning Representations, 2025.

---

### Note · Program_Chairs · 2026-01-17
**Submission Desk Rejected by Program Chairs**

The following references in this submission do not refer to real documents and/or have major errors in bibliographic information:

 Hao Wu, Xueyang Chen, Runlong Wang, Zicheng Yuan, Licheng Wang, Zhenggang Zhang, Jie Zhou, and Joseph J Lim. Visual programming for compositional reasoning and action planning in autonomous agents. arXiv preprint arXiv:2310.00425, 2023.